# Online Pre-Training for Offline-to-Online Reinforcement Learning

**Yongjae Shin**[2,1†] **Jeonghye Kim**[2,1†] **Whiyoung Jung**[1] **Sunghoon Hong**[1] **Deunsol Yoon**[1] **Youngsoo Jang**[1]
**Geonhyeong Kim**[1] **Jongseong Chae**[2] **Youngchul Sung**[2] **Kanghoon Lee**[1] **Woohyung Lim**[1]

## Abstract

Offline-to-online reinforcement learning (RL) aims to integrate the complementary strengths of offline and online RL by pre-training an agent offline and subsequently fine-tuning it through online interactions. However, recent studies reveal that offline pre-trained agents often underperform during online fine-tuning due to inaccurate value estimation caused by distribution shift, with random initialization proving more effective in certain cases. In this work, we propose a novel method, Online Pre-Training for Offline-to-Online RL (OPT), explicitly designed to address the issue of inaccurate value estimation in offline pre-trained agents. OPT introduces a new learning phase, Online Pre-Training, which allows the training of a new value function tailored specifically for effective online fine-tuning. Implementation of OPT on TD3 and SPOT demonstrates an average 30% improvement in performance across a wide range of D4RL environments, including MuJoCo, Antmaze, and Adroit.

## 1. Introduction

Reinforcement Learning (RL) has shown great potential in addressing complex decision-making tasks across various fields (Mnih et al. 2015; Silver et al. 2017). In particular, offline RL (Levine et al. 2020) offers the advantage of training an agent on the fixed dataset, thereby mitigating the potential costs or risks associated with direct interactions in real-world environments - a significant limitation of online RL. However, the effectiveness of offline RL is inherently constrained by the quality of the dataset, which can impede the overall performance.

†Work done during an internship at LG AI Research. [1]LG AI Research, Seoul, Republic of Korea. [2]School of Electrical Engineering, KAIST, Daejeon, Republic of Korea. Correspondence to: Woohyung Lim <w.lim@lgresearch.ai>.

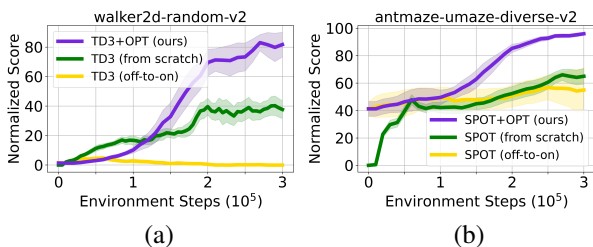

*Figure 1.* Comparison between offline-to-online RL (yellow), from scratch (green), and our method (purple). (a): A TD3+BC (Fujimoto & Gu 2021) pre-trained agent is fine-tuned with TD3 (Fujimoto et al. 2018). (b): A SPOT (Wu et al. 2022) pre-trained agent is fine-tuned with the same algorithm.

To overcome the cost challenge of online RL and the performance limitation of offline RL, the offline-to-online RL approach has been introduced (Lee et al. 2022; Zhang et al. 2023; Yu & Zhang 2023). This approach entails training an agent sufficiently on an offline dataset, followed by fine-tuning through additional interactions with the environment. The offline phase allows the agent to acquire a reasonable initial policy without any online cost, providing a stable foundation for subsequent fine-tuning. Combining the strengths of both approaches, offline-to-online RL reduces the need for extensive environment interactions, while enhancing the agent's performance through online fine-tuning.

Although offline-to-online RL offers clear advantages, prior studies (Zhang et al. 2023; Guo et al. 2023; Nakamoto et al. 2024; Zhang et al. 2024; Kong et al. 2024; Hu et al. 2024; Luo et al. 2024) have shown that fine-tuning an offline pre-trained agent often results in worse performance compared to training from scratch. This phenomenon, described by Nakamoto et al. (2024) as *counter-intuitive trends*, is depicted in Figure 1, which compares the learning curves during the online phase for both fine-tuning (`off-to-on`) and training from scratch with the replay buffer initialized using the offline dataset (`from scratch`). As shown in Figure 1 (a), training from scratch outperforms fine-tuning from the very beginning of the learning process, as also observed by Ball et al. (2023). In Figure 1 (b), although fine-tuning demonstrates moderate performance in the early stages, it is eventually surpassed by training from scratch. This highlights the inherent challenges in fine-tuning pre-trained agents in offline-to-online settings.

Previous studies (Nakamoto et al. 2024; Zhang et al. 2024) attribute this *counter-intuitive trends* of online fine-tuning to issues stemming from inaccurate value estimation. In response, Nakamoto et al. (2024) focuses on providing a lower bound for value updates, applied from the offline phase onward to correct value estimates throughout learning. Meanwhile, Zhang et al. (2024) introduces perturbations to value updates, aiming to stabilize learning by promoting smoother value estimation. One notable characteristic shared by these approaches is their reliance on the single value function across both the offline and online learning phases. In contrast, we adopt a different approach by introducing and utilizing an entirely new value function. To investigate this, we formulate two key research questions that guide our exploration:

Q1. *"Can adding a new value function resolve the issue of slow performance improvement?"*

Q2. *"How can we best leverage the new value function?"*

Through a comprehensive analysis of these research questions, we propose Online Pre-Training for Offline-to-Online RL (OPT), a novel approach that introduces a new value function to leverage it during online fine-tuning. In response to the first research question, OPT introduces a new value function, enhancing overall performance, as illustrated in Figure 1. Rather than relying solely on a value function trained in the offline phase, this additional value function allows the agent to better adapt to the changing dynamics of the online setting. For the second research question, OPT incorporates an additional learning phase, termed Online Pre-Training, which focuses on learning this new value function. This value function is then utilized during online fine-tuning to guide policy learning, ensuring more effective performance improvements.

OPT consistently demonstrates strong performance across various D4RL environments within a limited setting of 300k online interactions, surpassing previous state-of-the-art results. Additionally, since OPT introduces a new value function as one of its key components, it can be broadly integrated into existing value function-based RL algorithms. This design enables OPT to serve as a general enhancement module, compatible with a wide range of existing methods. We empirically verify this versatility through extensive experiments.

## 2. Background and Related Work

Reinforcement Learning (RL) is modeled as a Markov Decision Process (MDP) (Puterman 1990). In this framework, at each time step, an agent selects an action $a$ based on its current state $s$ according to its policy $\pi(a|s)$. The environment transitions to a subsequent state $s'$ and provides a reward

$r$, following the transition probability $p(s'|s, a)$ and reward function $r(s, a)$, respectively. Over successive interactions, the agent's policy $\pi$ is optimized to maximize the expected cumulative return $\mathbb{E}_\pi \left[ \sum_t \gamma^t r(s_t, a_t) \right]$, where $\gamma \in [0, 1)$ is the discount factor.

Offline RL focuses on training agents using a static dataset $D = \{(s, a, r, s')\}$, usually generated by various policies. To address the constraints of offline RL, which are often limited by the quality of the dataset, offline-to-online RL introduces an additional phase of online fine-tuning. In this framework, an offline pre-trained agent continues learning through online fine-tuning, enabling further improvement beyond what the offline dataset alone can offer. This hybrid paradigm allows the agent to adapt and improve by effectively leveraging new experiences collected during online interaction.

**Addressing Inaccurate Value Estimation.** Offline RL, reliant on a fixed dataset, is prone to extrapolation error when the value function evaluates out-of-distribution (OOD) actions (Kumar et al. 2020; Fujimoto et al. 2019; Kim et al. 2024). Several methods have been proposed to address this challenge: some focus on training the value function to assign lower values to OOD actions (Wu et al. 2019; Kostrikov et al. 2021a), while others aim to avoid OOD action evaluation altogether (Kostrikov et al. 2021b). These inaccurate value estimations in offline training not only degrade offline performance but can also adversely impact subsequent online fine-tuning (Yu & Zhang 2023; Zhang et al. 2024; Nakamoto et al. 2024; Kong et al. 2024; Feng et al. 2024; Luo et al. 2024; Zhou et al. 2025). Although constraints applied in offline RL can be extended to online fine-tuning to mitigate this issue (Kostrikov et al. 2021b; Lyu et al. 2022; Wu et al. 2022), such strategies often impose excessive conservatism, limiting the potential for performance enhancement.

Recent advances in offline-to-online RL have aimed to overcome this limitation arising from inaccurate value estimations (Nakamoto et al. 2024; Zhang et al. 2024; Luo et al. 2024). One approach (Nakamoto et al. 2024) addresses overconservatism during the offline phase by providing a lower bound for value updates. However, the conservative nature of this method continues to hinder policy improvement. Another approach (Zhang et al. 2024) introduces perturbations to value updates and increases their update frequency. While effective, this method incurs significantly higher computational costs compared to standard techniques, making it less practical for general use.

**Backbone Algorithms.** Our proposed Online Pre-Training process using newly introduced value function can be applied to various backbone algorithms. Among the many potential candidates, this study utilizes TD3+BC (Fujimoto & Gu 2021) and SPOT (Wu et al. 2022) as the backbone

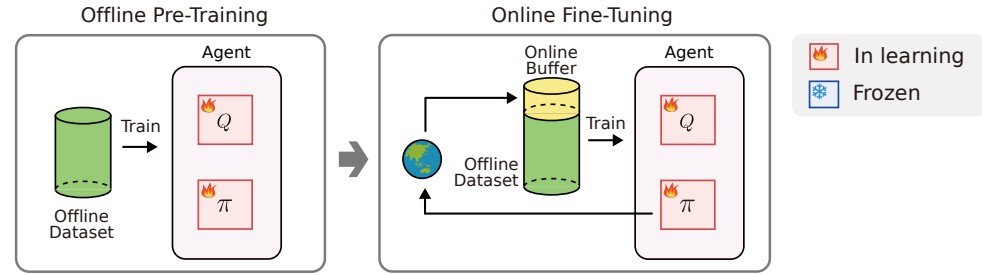

(a) Conventional Offline-to-Online RL

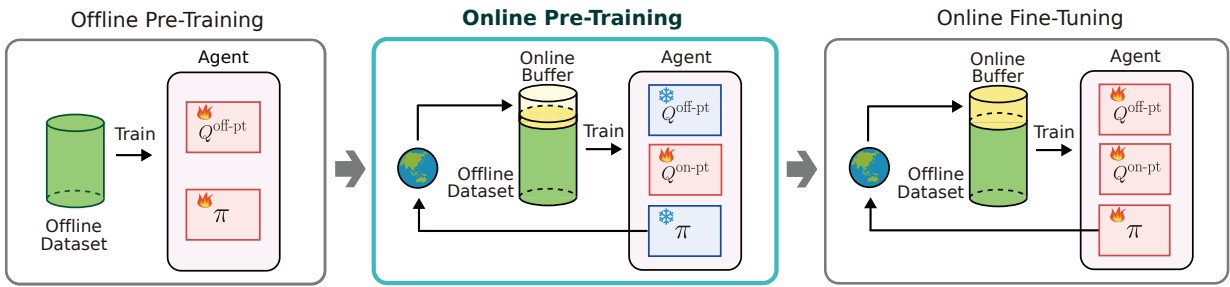

(b) **Ours:** Online Pre-Training for Offline-to-Online RL

*Figure 2.* Illustrations of two different learning methods: (a) Conventional Offline-to-Online RL (b) **Ours**. OPT introduces a new learning phase, termed Online Pre-Training, between offline pre-training and online fine-tuning. The illustration indicates whether the value function and policy are *in learning* or *frozen* (not being trained) during each training phase.

algorithms due to their simplicity and sample efficiency. TD3+BC extends the original TD3 (Fujimoto et al. 2018) algorithm by incorporating a behavior cloning (BC) regularization term into the policy improvement. The value function is trained using temporal-difference (TD) learning, with the loss functions for both the policy $\pi_\phi(s)$ and value function $Q_\theta(s, a)$ defined as follows:

$$\mathcal{L}_\pi(\phi) = \mathbb{E}_{s \sim B}[-Q_\theta(s, \pi_\phi(s)) + \alpha(\pi_\phi(s) - a)^2], \quad (1)$$

$$\mathcal{L}_Q(\theta) = \mathbb{E}_{(s,a,r,s') \sim B}[(Q_\theta(s, a) \\ -(r + \gamma Q_{\bar{\theta}}(s', \pi_\phi(s'))))^2] \quad (2)$$

where $Q_{\bar{\theta}}$ denotes a delayed target value function and $B$ is the replay buffer. SPOT extends of TD3+BC by replacing the BC regularization term with a pre-trained VAE, which is then used to penalize OOD actions based on the uncertainty.

## 3. Method

In this section, we introduce Online Pre-Training for Offline-to-Online RL (OPT), a novel method aimed at addressing the issue of inaccurate value estimation in offline-to-online RL. The proposed method employs two value functions, $Q^{\text{off-pt}}$ and $Q^{\text{on-pt}}$, each serving distinct roles in the time domain. The method consists of three distinct stages of learning:

(i) Offline Pre-Training: Following the conventional offline-to-online RL, the offline pre-trained value function $Q^{\text{off-pt}}$ and policy $\pi^{\text{off}}$ are jointly trained on the offline dataset.

(ii) Online Pre-Training: The newly initialized value function, $Q^{\text{on-pt}}$, is trained on both the offline dataset and newly collected online samples.

(iii) Online Fine-Tuning: The policy is updated by utilizing both $Q^{\text{off-pt}}$ and $Q^{\text{on-pt}}$, with each value function continuously updated.

Figure 2 illustrates the difference between conventional offline-to-online learning (Figure 2a) and our proposed method, OPT (Figure 2b). OPT introduces a new learning phase called Online Pre-Training, making it distinct from the conventional two-stage offline-to-online RL methods by comprising three stages. The following sections focus on the Online Pre-Training and Online Fine-Tuning phases, while the offline pre-training phase adheres to the standard offline RL process.

### 3.1. Online Pre-Training

In the proposed method, $Q^{\text{on-pt}}$ is introduced as an additional value function specifically designed for online fine-tuning.

Considering $Q^{\text{on-pt}}$, one straightforward approach is to add a randomly initialized value function. In this case, as $Q^{\text{on-pt}}$ begins learning from the online fine-tuning, it is expected to adapt well to the new data encountered during online fine-tuning. However, since $Q^{\text{on-pt}}$ is required to train from scratch, it often disrupts policy learning in the early stages. To address the potential negative effects that can arise from adding $Q^{\text{on-pt}}$ with random initialization, we introduce a pre-training phase, termed Online Pre-Training, specifically designed to train $Q^{\text{on-pt}}$ in advance of online fine-tuning. The following sections explore the design of the Online Pre-Training method in detail.

**Designing Datasets.** As the initial stage of Online Pre-Training, the only available dataset to train $Q^{\text{on-pt}}$ is the offline dataset $\mathcal{B}_{\text{off}}$. Yet, given that $B_{\text{off}}$ has already been extensively utilized to train $Q^{\text{off-pt}}$, further relying solely on this dataset inevitably causes $Q^{\text{on-pt}}$ to closely replicate $Q^{\text{off-pt}}$. To address this, we incorporate online samples by initiating Online Pre-Training with the collection of $N_\tau$ samples in the online buffer $B_{\text{on}}$, generated by $\pi^{\text{off}}$. To leverage $B_{\text{on}}$, one approach is to train $Q^{\text{on-pt}}$ with $B_{\text{on}}$. Since $Q^{\text{on-pt}}$ is trained based on $\pi^{\text{off}}$, this prevents $Q^{\text{on-pt}}$ from disrupting policy learning in the initial stage of online fine-tuning. However, as $B_{\text{on}}$ is generated by the fixed policy $\pi^{\text{off}}$, relying solely on $B_{\text{on}}$ risks limiting the generalizability of $Q^{\text{on-pt}}$ to diverse online samples. Therefore, to address both issue of similarity to $Q^{\text{off-pt}}$ and the risk of overfitting to $\pi^{\text{off}}$, a balanced approach utilizing both $B_{\text{off}}$ and $B_{\text{on}}$ is necessary during training.

**Designing Objective Function.** By leveraging both datasets, the objective for $Q^{\text{on-pt}}$ is to ensure its adaptability to the evolving policy samples, promoting continuous policy improvement and enhancing sample efficiency. To achieve this, we propose a meta-adaptation objective inspired by the OEMA (Guo et al. 2023), tailored to our setting. The objective function for $Q^{\text{on-pt}}$ is outlined as follows:

$$\mathcal{L}_{Q^{\text{on-pt}}}^{\text{pretrain}}(\psi) = \mathcal{L}_{Q^{\text{on-pt}}}^{\text{off}}(\psi) + \mathcal{L}_{Q^{\text{on-pt}}}^{\text{on}}(\psi - \alpha \nabla \mathcal{L}_{Q^{\text{on-pt}}}^{\text{off}}(\psi))$$
(3)

where $\quad \mathcal{L}_{Q^{\text{on-pt}}}^{\text{off}}(\psi) = \mathbb{E}_{s,a,r,s' \in B_{\text{off}}}[(Q_\psi^{\text{on-pt}}(s,a)$
$$- (r + \gamma Q_{\bar{\psi}}^{\text{on-pt}}(s', \pi_\phi(s'))))^2],$$
$$\mathcal{L}_{Q^{\text{on-pt}}}^{\text{on}}(\psi) = \mathbb{E}_{s,a,r,s' \in B_{\text{on}}}[(Q_\psi^{\text{on-pt}}(s,a)$$
$$- (r + \gamma Q_{\bar{\psi}}^{\text{on-pt}}(s', \pi_\phi(s'))))^2].$$

Here, $Q_{\bar{\psi}}^{\text{on-pt}}$ represents the target network. Equation (3) consists of two terms: the first term facilitates learning from $B_{\text{off}}$, while the second term serves as an objective to ensure that $Q^{\text{on-pt}}$ adapts to $B_{\text{on}}$. Optimizing these terms allows $Q^{\text{on-pt}}$ to leverage $B_{\text{off}}$ and $B_{\text{on}}$ to align closely with the dynamics of the current policy, thereby enabling efficient adaptation to online samples during fine-tuning. During

---

**Algorithm 1** OPT: Online Pre-Training for Offline-to-Online Reinforcement Learning

1: **Inputs:** Offline dataset $\mathcal{B}_{\text{off}}$, offline trained agent $\{Q_\theta^{\text{off-pt}}, \pi_\phi\}$, online pre-training samples $N_\tau$, online pre-training steps $N_{\text{pretrain}}$, online fine-tuning steps $N_{\text{finetune}}$, weighting coefficient $\kappa$
2: Initialize online replay buffer $\mathcal{B}_{\text{on}}$, value function $Q_\psi^{\text{on-pt}}$
   `// Online Pre-Training`
3: Store $N_\tau$ transitions $\tau = (s, a, r, s')$ in $\mathcal{B}_{\text{on}}$ via environment interaction with $\pi_\phi$
4: **for** $i = 1$ **to** $N_{\text{pretrain}}$ **do**
5: $\quad$ Sample minibatch of transitions $\{\tau_j\}_{j=1}^B \sim \mathcal{B}_{\text{off}}$, $\{\tau_j\}_{j=1}^B \sim \mathcal{B}_{\text{on}}$
6: $\quad$ Update $\psi$ minimizing $\mathcal{L}_{Q^{\text{on-pt}}}^{\text{pretrain}}(\psi)$ by Equation (3)
7: **end for**
   `// Online Fine-Tuning`
8: Initialize balanced replay buffer $\mathcal{B}_{\text{BR}} \leftarrow \mathcal{B}_{\text{off}} \cup \mathcal{B}_{\text{on}}$
9: **for** $i = 1$ **to** $N_{\text{finetune}}$ **do**
10: $\quad$ Sample minibatch of transitions $\tau \sim \mathcal{B}_{\text{BR}}$
11: $\quad$ Update $\theta$ and $\psi$ minimizing $\mathcal{L}_{Q^{\text{off-pt}}}(\theta)$ and $\mathcal{L}_{Q^{\text{on-pt}}}(\psi)$ respectively by Equation (2)
12: $\quad$ Update $\phi$ minimizing $\mathcal{L}_\pi^{\text{finetune}}(\phi)$ by Equation (4)
13: **end for**

---

the Online Pre-Training, only $Q^{\text{on-pt}}$ is updated, with no alterations made to other components, $\pi^{\text{off}}$ and $Q^{\text{off-pt}}$.

### 3.2. Online Fine-Tuning

As $Q^{\text{on-pt}}$ is trained during Online Pre-Training to effectively adapt to online samples, we leverage it in the online fine-tuning to guide policy learning. Throughout this phase, the buffer $B_{\text{on}}$ is continuously filled and the learning of all three components $Q^{\text{off-pt}}$, $Q^{\text{on-pt}}$ and $\pi_\phi$ progresses. As in the conventional offline-to-online RL, $Q^{\text{off-pt}}$ is updated using TD learning, and similarly, $Q^{\text{on-pt}}$, which was trained by Equation (3) during Online Pre-Training, is also updated via TD learning.

**Effectively Balancing $Q^{\text{off-pt}}$ and $Q^{\text{on-pt}}$.** One of the key aspects of OPT lies in its policy improvement strategy, which, unlike most previous approaches that rely on a single value function, effectively balances both $Q^{\text{off-pt}}$ and $Q^{\text{on-pt}}$ for policy improvement. Since $Q^{\text{off-pt}}$ provides reliable information about the offline dataset, and $Q^{\text{on-pt}}$, trained via a meta-adaptation strategy during Online Pre-Training, is well-suited for adapting to newly collected data, we propose a policy learning strategy that effectively exploits the complementary strengths of both value functions. The proposed loss function for policy improvement is given by:

$$\mathcal{L}_\pi^{\text{finetune}}(\phi) = \mathbb{E}_{s \sim B}[-\{(1 - \kappa)Q^{\text{off-pt}}(s, \pi_\phi(s))$$
$$+ \kappa Q^{\text{on-pt}}(s, \pi_\phi(s))\}],$$
(4)

*Table 1.* Comparison of the normalized scores after online fine-tuning for each environment in MuJoCo domain. r = random, m = medium, m-r = medium-replay. All results are reported as the mean and 95% confidence interval across ten random seeds.

| Environment | TD3 | Off2On | OEMA | PEX | ACA | FamO2O | Cal-QL | TD3 + OPT (Ours) |
|---|---|---|---|---|---|---|---|---|
| halfcheetah-r | **94.6**$_{\pm 2.7}$ | **92.8**$_{\pm 3.5}$ | 83.8$_{\pm 4.7}$ | 62.3$_{\pm 3.5}$ | 91.0$_{\pm 1.6}$ | 38.7$_{\pm 2.4}$ | 32.2$_{\pm 4.5}$ | 90.2$_{\pm 1.6}$ |
| hopper-r | 86.0$_{\pm 13.3}$ | 94.5$_{\pm 5.2}$ | 59.8$_{\pm 13.4}$ | 46.5$_{\pm 22.4}$ | 84.6$_{\pm 12.5}$ | 12.0$_{\pm 1.3}$ | 10.3$_{\pm 11.7}$ | **108.7**$_{\pm 1.7}$ |
| walker2d-r | 0.1$_{\pm 0.1}$ | 29.4$_{\pm 2.9}$ | 16.7$_{\pm 7.1}$ | 10.7$_{\pm 1.8}$ | 38.8$_{\pm 15.4}$ | 9.9$_{\pm 0.5}$ | 10.9$_{\pm 2.7}$ | **88.0**$_{\pm 3.2}$ |
| halfcheetah-m | 93.4$_{\pm 1.9}$ | **103.3**$_{\pm 0.7}$ | 67.6$_{\pm 11.6}$ | 69.4$_{\pm 3.7}$ | 81.0$_{\pm 0.6}$ | 49.8$_{\pm 0.2}$ | 69.9$_{\pm 5.2}$ | 97.0$_{\pm 0.8}$ |
| hopper-m | 89.3$_{\pm 14.5}$ | 108.4$_{\pm 1.4}$ | 107.1$_{\pm 0.9}$ | 84.4$_{\pm 13.3}$ | 101.6$_{\pm 2.1}$ | 74.4$_{\pm 5.3}$ | 102.3$_{\pm 1.1}$ | **112.2**$_{\pm 0.6}$ |
| walker2d-m | 103.5$_{\pm 3.8}$ | **112.3**$_{\pm 12.8}$ | 98.9$_{\pm 4.1}$ | 84.5$_{\pm 10.9}$ | 79.9$_{\pm 12.4}$ | 84.4$_{\pm 1.4}$ | 96.1$_{\pm 3.5}$ | **116.1**$_{\pm 1.9}$ |
| halfcheetah-m-r | 87.2$_{\pm 0.9}$ | **97.7**$_{\pm 1.7}$ | 48.5$_{\pm 11.8}$ | 55.9$_{\pm 0.8}$ | 67.6$_{\pm 2.0}$ | 48.1$_{\pm 0.3}$ | 64.8$_{\pm 1.9}$ | 94.3$_{\pm 1.5}$ |
| hopper-m-r | 94.5$_{\pm 11.3}$ | 103.7$_{\pm 6.7}$ | 108.8$_{\pm 0.8}$ | 85.8$_{\pm 11.3}$ | 104.7$_{\pm 2.2}$ | 98.5$_{\pm 3.9}$ | 101.8$_{\pm 2.1}$ | **112.7**$_{\pm 0.4}$ |
| walker2d-m-r | 103.9$_{\pm 6.6}$ | **118.8**$_{\pm 4.6}$ | 103.4$_{\pm 2.9}$ | 89.3$_{\pm 6.3}$ | 89.3$_{\pm 14.0}$ | 88.6$_{\pm 6.6}$ | 98.5$_{\pm 1.4}$ | **119.9**$_{\pm 2.2}$ |
| Total | 752.4 | 860.9 | 694.6 | 588.8 | 738.5 | 504.4 | 586.8 | **939.1** |

*Table 2.* Comparison of the normalized scores after online fine-tuning for each environment in Antmaze domain. All results are reported as the mean and 95% confidence interval across ten random seeds.

| Environment | SPOT | PEX | ACA | FamO2O | Cal-QL | SPOT + OPT (Ours) |
|---|---|---|---|---|---|---|
| umaze | 98.7$_{\pm 0.8}$ | 95.6$_{\pm 0.8}$ | 95.0$_{\pm 1.7}$ | 93.0$_{\pm 1.6}$ | 90.8$_{\pm 5.9}$ | **99.7**$_{\pm 0.2}$ |
| umaze-diverse | 55.9$_{\pm 15.9}$ | 30.7$_{\pm 17.7}$ | 94.5$_{\pm 1.4}$ | 43.5$_{\pm 14.1}$ | 75.2$_{\pm 15.9}$ | **97.7**$_{\pm 0.4}$ |
| medium-play | 91.1$_{\pm 2.3}$ | 85.0$_{\pm 2.1}$ | 0.0$_{\pm 0.0}$ | 89.3$_{\pm 2.1}$ | 94.6$_{\pm 2.9}$ | **97.6**$_{\pm 0.8}$ |
| medium-diverse | 92.0$_{\pm 1.7}$ | 85.4$_{\pm 2.1}$ | 0.0$_{\pm 0.0}$ | 77.9$_{\pm 15.8}$ | 96.2$_{\pm 2.0}$ | **98.7**$_{\pm 0.6}$ |
| large-play | 58.0$_{\pm 11.4}$ | 57.5$_{\pm 3.6}$ | 0.0$_{\pm 0.0}$ | 55.8$_{\pm 5.0}$ | 73.7$_{\pm 6.8}$ | **81.5**$_{\pm 4.5}$ |
| large-diverse | 70.0$_{\pm 11.5}$ | 58.3$_{\pm 4.7}$ | 0.0$_{\pm 0.0}$ | 56.8$_{\pm 4.4}$ | 72.9$_{\pm 7.8}$ | **90.1**$_{\pm 2.9}$ |
| Total | 465.7 | 412.5 | 189.5 | 416.3 | 503.4 | **565.3** |

where $\pi_\phi$ is initialized as $\pi^{\text{off}}$ and $0 < \kappa \leq 1$. $\kappa$ is a weighting coefficient that regulates the balance between $Q^{\text{off-pt}}$ and $Q^{\text{on-pt}}$. When the discrepancy between the offline dataset and newly collected online samples is small, $Q^{\text{off-pt}}$ remains reliable, making it suitable for guiding policy improvement. Accordingly, a smaller $\kappa$ values applied to more effectively leverage $Q^{\text{off-pt}}$. As online fine-tuning progresses, $Q^{\text{on-pt}}$, optimized through our meta-adaptation objective during Online Pre-Training, quickly adapts to the online data. To leverage this adaptability, $\kappa$ is incrementally increased to shift the reliance towards the more rapidly adapting $Q^{\text{on-pt}}$. In summary, as online fine-tuning progresses, we schedule $\kappa$ to progressively shift the influence on policy learning from $Q^{\text{off-pt}}$ to $Q^{\text{on-pt}}$. This dynamic adjustment ensures that the policy is always guided by the most informative value estimate at each stage of training.

Additionally, to promote the use of online samples, we employ balanced replay (Lee et al. 2022), which prioritizes samples encountered during online interactions to further accelerate the adaptation of $Q^{\text{on-pt}}$. The overall learning phase of OPT is illustrated in Figure 2b, with the algorithm presented in Algorithm 1. The implementation details are presented in Appendix A.1, and the hyperparameter settings are summarized in Appendix H.

# 4. Experiments

In this section, we demonstrate the effectiveness of our proposed method through experimental results. In Section 4.1, we describe our experimental setup and compare our method against existing offline-to-online RL approaches across various environments. Section 4.2 examines the applicability and performance of our method when applied to an alternative value-based algorithm.

## 4.1. Main Results

**Experimental Setup.** We evaluate the performance of OPT across three domains from the D4RL benchmark (Fu et al. 2020). MuJoCo is a suite of locomotion tasks including datasets of diverse quality for each environment. Antmaze is a set of navigation tasks where an ant robot is controlled to navigate from a starting point to a goal location within a maze. Adroit is a set of robot manipulation tasks that require controlling a five-finger robotic hand to achieve a specific goal in each task. A detailed description of the environment and dataset is provided in Appendix B. For all baselines, the offline phase comprises 1M gradient steps, and the online phase consists of 300k environment steps. OPT carries out Online Pre-Training for the first 25k steps of the online phase, followed by online fine-tuning for the

*Table 3.* Comparison of the normalized scores after online fine-tuning for each environment in Adroit domain. All results are reported as the mean and 95% confidence interval across ten random seeds.

| Environment | SPOT | Cal-QL | SPOT + OPT (Ours) |
|---|---|---|---|
| pen-cloned | $114.8_{\pm6.3}$ | $0.21_{\pm3.2}$ | $\mathbf{136.2}_{\pm4.8}$ |
| hammer-cloned | $84.0_{\pm13.8}$ | $0.23_{\pm0.03}$ | $\mathbf{121.9}_{\pm1.6}$ |
| door-cloned | $1.6_{\pm2.7}$ | $-0.33_{\pm0.00}$ | $\mathbf{51.1}_{\pm12.9}$ |
| relocate-cloned | $-0.19_{\pm0.02}$ | $-0.34_{\pm0.00}$ | $\mathbf{-0.06}_{\pm0.03}$ |
| Total | 200.2 | -0.23 | **309.1** |

remaining 275k steps thus, like the other baselines, it also has a total 300k environment steps in the online phase. The implementation details are provided in Appendix A.1.

**Baselines.** We compare OPT with the following baselines: (1) Off2On (Lee et al. 2022), an ensemble-based method that incorporates balanced replay to promote the use of near-on-policy samples; (2) OEMA (Guo et al. 2023), which applies an optimistic exploration strategy alongside a meta-adaptation method for policy learning; (3) PEX (Zhang et al. 2023), which utilizes a set of policies, including a frozen offline pre-trained policy and an additional learnable policy, designed to balance exploration and exploitation during online fine-tuning; (4) ACA (Yu & Zhang 2023), which post-processes the offline pre-trained value function to align it with the policy, where the post-processing is specifically applied at the transition point between offline pre-training and online fine-tuning; (5) FamO2O (Wang et al. 2024), which employs a state-adaptive policy improvement method by leveraging both a balance model and a universal model to generalize across diverse state distributions; and (6) Cal-QL (Nakamoto et al. 2024), which trains a value function with an explicit lower bound constraint to mitigate over-conservatism introduced during the offline pre-training phase.

**Results on MuJoCo.** In the MuJoCo, we implement OPT on the baseline which utilizes TD3+BC for the offline phase, followed by TD3 in the online phase. To further improve sample efficiency during online fine-tuning, we adopt an update-to-data (UTD) ratio of 5 for TD3+OPT and apply the same setting to TD3 to ensure a fair comparison. The results in Table 1 indicate that OPT demonstrates strong overall performance, notably surpassing the existing state-of-the-art (SOTA) in several environments. Its consistently high scores across different datasets highlight the robustness of OPT in handling diverse dynamics and data distributions. In particular, the results for the walker2d-random-v2 dataset demonstrate the remarkable efficacy of OPT, as it significantly outperforms existing approaches.

**Results on Antmaze.** In the Antmaze, we implement OPT within the SPOT, as TD3+BC showed suboptimal

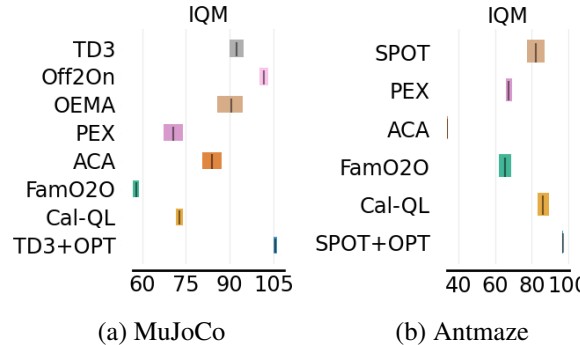

(a) MuJoCo                (b) Antmaze

*Figure 3.* Interquartile Mean (IQM) comparison of baseline methods on the MuJoCo and Antmaze domains. The x-axis represents normalized scores. All results are based on 10 random seeds.

performance in this domain. Table 2 shows that OPT consistently delivers superior performance across all environments. When comparing to Cal-QL (Nakamoto et al. 2024), a recently proposed method with the same objective of addressing inaccurate value estimation, the results in the umaze-diverse and large-diverse environments demonstrate the efficacy of introducing a new value function through Online Pre-Training in mitigating this issue.

**Results on Adroit.** In the Adroit, as with Antmaze, we apply OPT to the SPOT. Table 3 demonstrates that, unlike Cal-QL (Nakamoto et al. 2024), which struggles to learn from low-quality datasets such as cloned due to its conservative nature, OPT manages to perform well even with these challenging datasets. In particular, the results in the door-cloned environment, where SPOT fails to perform adequately, demonstrate the effectiveness of OPT.

**Overall Findings** The results across all domains indicate that OPT consistently achieves strong performance, demonstrating its versatility and effectiveness. Notably, while Off2On (Lee et al. 2022) performs competitively in the MuJoCo domain, prior studies (Li et al. 2023; Lei et al. 2023) have mentioned its struggles in Antmaze-large and Adroit. In contrast, OPT demonstrates superior performance in Antmaze and Adroit, underscoring its robustness and ability to generalize across diverse domains and dataset conditions. The individual learning curves can be found in Appendix I.

To further support the statistical significance of our results, we report Interquartile Mean (IQM) (Agarwal et al. 2021) scores, which offer a more reliable metric by averaging performance within the 25th to 75th percentile. IQM reduces the influence of extreme outliers and better captures consistent trends across tasks. As shown in Figure 3, OPT achieves superior performance compared to other baselines, with non-overlapping 95% confidence intervals, thereby demonstrating the statistical robustness of our results. These findings provide additional evidence that OPT not only per-

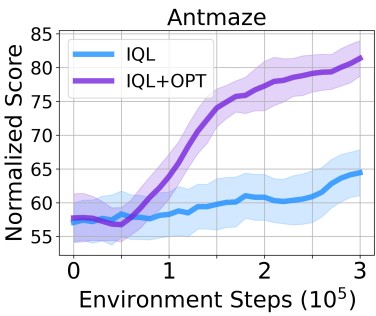

*Figure 4.* Aggregated return curves for IQL and IQL+OPT, averaged across all six environments in the Antmaze domain.

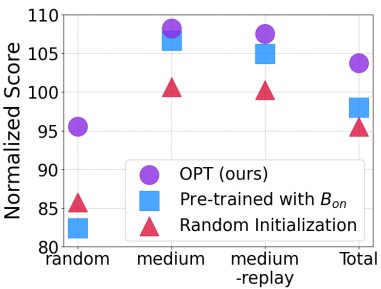

*Figure 5.* Comparing normalized score of OPT with different initialization methods, averaging across all 3 environments in the MuJoCo domain.

forms well on average but also does so consistently across runs and environments.

### 4.2. Extending OPT to IQL

So far, we have shown that OPT is an effective algorithm for online fine-tuning when applied to TD3-based algorithms, such as TD3 and SPOT. To further validate the versatility of OPT, we changed the backbone algorithm to IQL (Kostrikov et al. 2021b). This transition allows us to examine whether OPT remains effective when applied to algorithms with fundamentally different policy and value function structures. Unlike TD3-based algorithms, which employ a deterministic policy and rely solely on a state-action value function, IQL employs a stochastic policy and uses both a state-action value function and a state value function. Accordingly, to implement OPT, we modified the training loss, with detailed explanations provided in Appendix A.2.

The results in Figure 4 show that applying OPT yields a clear and noticeable performance improvement compared to the backbone algorithm. Notably, while IQL exhibits a gradual and steady performance increase, IQL+OPT demonstrates a more rapid and substantial improvement. These results indicate that OPT is not limited to a specific backbone algorithm but can be effectively integrated into various value-based methods, serving as a general strategy for enhancing performance. Additional experiment results in different domains, including MuJoCo and Adroit, are presented in Appendix E.

## 5. Discussion

In this section, we provide a detailed analysis of key components of OPT. Section 5.1 investigates the effect of different initialization strategies for $Q^{\text{on-pt}}$. Section 5.2 explores how the characteristics of offline datasets influence the scheduling of $\kappa$. Section 5.3 examines the role of the additional value function. Section 5.4 examines the impact of the $\kappa$ scheduling strategy. Section 5.5 focuses on the effect of the number of Online Pre-Training samples ($N_\tau$). Finally,

Section 5.6 presents a comparison between OPT and RLPD. Additional experiments and analyses are provided in Appendix C and Appendix D.

### 5.1. Effect of Initialization Methods for $Q^{\text{on-pt}}$

In Section 3.1, to best leverage $Q^{\text{on-pt}}$ during online fine-tuning, we explore various approaches for initializing $Q^{\text{on-pt}}$. To further substantiate the effectiveness of our initialization method through experimental results, we evaluate three approaches for initializing $Q^{\text{on-pt}}$: (1) Random Initialization, (2) Pre-trained with $B_{\text{on}}$, and (3) our proposed method, OPT. (1) Random Initialization initializes $Q^{\text{on-pt}}$ from scratch and uses it directly at the beginning of online fine-tuning, without any Online Pre-Training. (2) Pre-trained with $B_{\text{on}}$ trains $Q^{\text{on-pt}}$ during the Online Pre-Training using only samples from $B_{\text{on}}$ and the loss objective $\mathcal{L}^{\text{on}}_{Q^{\text{on-pt}}}(\psi)$ from Equation (3), and then uses it for online fine-tuning.

Figure 5 presents the average performance of the three initialization approaches across different datasets. It illustrates that random initialization consistently underperforms compared to OPT. This discrepancy, as discussed in Section 3.1, can be attributed to the fact that random initialization hinders policy learning from the outset, leading to suboptimal value estimates and ultimately degraded performance. Similarly, the results for pre-training with $B_{\text{on}}$ demonstrate that learning solely from $B_{\text{on}}$ is insufficient to follow the performance of OPT. In particular, the random dataset, where the policy evolves more drastically (Figure 6), highlights that an overfitted $Q^{\text{on-pt}}$, trained only on $B_{\text{on}}$, not only fails to adapt to this policy improvement but in some cases even performs worse than random initialization. This underscores the importance of both $B_{\text{off}}$ and $B_{\text{on}}$ for training. Detailed experimental results, including an additional approach and results on the Antmaze domain, are provided in Appendix C.3.

### 5.2. Influence of Dataset Characteristics on $\kappa$ Scheduling

In our proposed method, we utilize $\kappa$ to balance the contribution of $Q^{\text{off-pt}}$ and $Q^{\text{on-pt}}$ during online fine-tuning. To

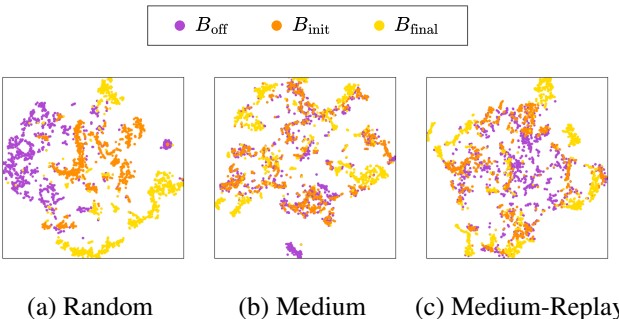

(a) Random  (b) Medium  (c) Medium-Replay

*Figure 6.* A t-SNE visualization of the offline dataset ($B_{\text{off}}$) and the rollout samples of the policy at the beginning ($B_{\text{init}}$) and end ($B_{\text{final}}$) of the online fine-tuning.

effectively leverage the strengths of both value functions, we adjust the $\kappa$ schedule based on the quality of the dataset, as different datasets offer varying levels of reliability for each value function. To provide insight into the rationale behind the scheduling of $\kappa$, we visualize the distributional differences between the offline dataset and policy-generated samples. Specifically, we compare the state-action distributions of the offline dataset and those of the policy rollouts using t-SNE (Van der Maaten & Hinton 2008) in the walker2d environment.

Figure 6 shows the comparison of the distribution between the offline dataset ($B_{\text{off}}$) and the rollout samples of policy at both the beginning ($B_{\text{init}}$) and the end ($B_{\text{final}}$) of the online fine-tuning phase. For the medium and medium-replay datasets, we observe that the distributions are similar at the start of the online fine-tuning but diverse towards the end. In contrast, for the random dataset, a difference between the two distributions is evident from the beginning. For the medium and medium-replay datasets, since $Q^{\text{off-pt}}$ is informative for the offline dataset due to offline pre-training, we initially assign it a higher weight during the early stages of online fine-tuning. As the online fine-tuning progresses, the weight is gradually shifted toward $Q^{\text{on-pt}}$. However, for the random dataset, due to the substantial distribution difference, we primarily rely on $Q^{\text{on-pt}}$ from the start of online fine-tuning. We present additional experiments related to the sensitivity of $\kappa$ in Appendix C.2 and provide the specific values of $\kappa$ in Appendix H.

### 5.3. Contribution of the Additional Value Function

A key component of our proposed algorithm is the introduction of a new value function, which is trained during the Online Pre-Training phase and subsequently utilized in online fine-tuning. To evaluate the impact of this addition, we examine the performance when the new value function is excluded. Specifically, we assess the outcome where $Q^{\text{off-pt}}$ is trained during Online Pre-Training using the objective in Equation (3), without introducing $Q^{\text{on-pt}}$ (denoted as w/o $Q^{\text{on-pt}}$ in Figure 7). Under this setup, policy improvement

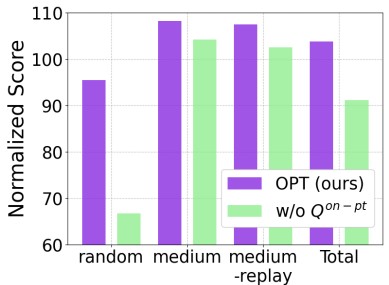

*Figure 7.* Comparison of the normalized score for OPT and its ablation (without the addition of a new value function), averaged across all 3 environments in the MuJoCo domain.

*Table 4.* Ablation study results on $\kappa$ in the Antmaze domain. All values are reported as the mean and 95% confidence interval over 5 random seeds.

| Environment | $0.1 \rightarrow 0.9$ | $0.1 \rightarrow 0.8$ | $0.1 \rightarrow 0.7$ | $0.5$ |
|---|---|---|---|---|
| umaze | $99.8_{\pm 0.3}$ | $100.0_{\pm 0.0}$ | $99.6_{\pm 0.7}$ | $99.0_{\pm 0.6}$ |
| umaze-diverse | $97.4_{\pm 0.3}$ | $96.8_{\pm 1.2}$ | $95.2_{\pm 3.2}$ | $97.0_{\pm 2.1}$ |
| medium-play | $98.2_{\pm 1.1}$ | $97.6_{\pm 1.4}$ | $98.2_{\pm 1.1}$ | $96.0_{\pm 1.8}$ |
| medium-diverse | $98.4_{\pm 1.1}$ | $97.6_{\pm 1.3}$ | $96.6_{\pm 2.1}$ | $97.0_{\pm 1.3}$ |
| large-play | $78.2_{\pm 3.8}$ | $78.3_{\pm 3.9}$ | $81.0_{\pm 6.4}$ | $59.2_{\pm 25.2}$ |
| large-diverse | $90.6_{\pm 3.2}$ | $90.0_{\pm 1.3}$ | $88.4_{\pm 3.2}$ | $75.4_{\pm 8.4}$ |

in online fine-tuning is driven solely by $Q^{\text{off-pt}}$.

The results presented in Figure 7 indicate that the addition of a new value function leads to improved performance regardless of the dataset. Notably, this improvement is pronounced in the random dataset. Due to the characteristics of the random dataset, where contains few successful demonstrations, $Q^{\text{off-pt}}$ becomes heavily biased. As a result, it fails to benefit from Online Pre-Training and performs poorly during online fine-tuning. Detailed experiment results are provided in Appendix C.5.

### 5.4. Impact of the $\kappa$ Scheduling Strategy

To effectively leverage both $Q^{\text{off-pt}}$ and $Q^{\text{on-pt}}$ during online fine-tuning, Section 3.2 introduces a weighted combination of the two value functions, controlled by a scheduling coefficient $\kappa$. The value of $\kappa$ is scheduled to change throughout online fine-tuning, gradually shifting the weighting from $Q^{\text{off-pt}}$ to $Q^{\text{on-pt}}$. To evaluate the impact of this scheduling strategy, we compare our proposed dynamic schedule (from $\kappa = 0.1$ to $0.9$) with a baseline using a fixed weighting of $\kappa = 0.5$. We also investigate the sensitivity of performance to the final value of $\kappa$ by testing alternative schedules with different endpoints.

As shown in Table 4, the use of scheduling leads to improved performance in relatively challenging environments such as Antmaze-large. This improvement suggests that gradually shifting supervision allows the agent to better adapt to online data distributions while maintaining stability early in

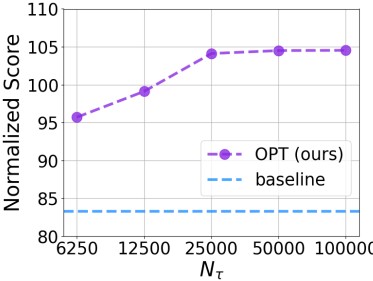

*Figure 8.* Comparing normalized score with varying $N_\tau$ of OPT, averaged across all 9 environments (3 tasks with 3 datasets each) in the MuJoCo domain.

training. Notably, the results suggest that performance is not particularly sensitive to the exact value of $\kappa$; rather, the key factor is the presence of a gradual transition from $Q^{\text{off-pt}}$ to $Q^{\text{on-pt}}$. Additional results using fixed $\kappa$ values of $\kappa = 0$ and $\kappa = 1$ are provided in Appendix C.

### 5.5. Effect of Online Pre-Training Sample Size ($N_\tau$)

The Online Pre-Training phase involves two hyperparameters: $N_\tau$ and $N_{\text{pretrain}}$. In particular, $N_\tau$ represents interactions with the environment used to collect online samples for Online Pre-Training. Since $N_\tau$ is also part of the environment steps within the online phase, determining the most efficient value for $N_\tau$ is critical for sample efficiency in offline-to-online RL. To investigate this, we conduct experiments in the MuJoCo domain comparing the results when $N_\tau$ is set to 1/4, 1/2, 2, and 4 times its current value (25k). In these experiments, $N_{\text{pretrain}}$ is set to twice $N_\tau$, consistent with the setting proposed in our method, while the number of online fine-tuning steps is kept constant.

Figure 12 demonstrates that increasing $N_\tau$ enhances the effectiveness of Online Pre-Training. However, beyond a certain threshold, further increases do not lead to additional performance gains. During environment interactions for $N_\tau$, the policy remained fixed, and once online data surpasses an optimal level, it no longer contributes to Online Pre-Training. Since offline-to-online RL aims for high performance with minimal interaction, these results suggest OPT is most efficient when $N_\tau$ is set to 25k. We adopt $N_\tau = 25k$ and $N_{\text{pretrain}} = 50k$ across all domains. Appendix C.4 presents detailed per-environment results.

### 5.6. Comparison with RLPD under Offline-to-Online Setting

Thus far, we proposed a method to integrate offline pretrained agents into online fine-tuning effectively. Recently, several studies have emerged demonstrating strong performance using online RL alone with offline datasets, without requiring an explicit offline phase (Song et al. 2022; Ball et al. 2023). Among these, RLPD (Ball et al. 2023) has

*Table 5.* Normalized score for each environment on the MuJoCo domain. The full results for other domains, including Antmaze and Adroit, are provided in Appendix F.

| Environment | RLPD | | |
|---|---|---|---|
| | Vanilla | Off-to-On | OPT (Ours) |
| halfcheetah-r | 91.5 ±2.5 | **96.1** ±5.2 | 90.7 ±2.2 |
| hopper-r | 90.2 ±19.1 | 95.7 ±18.4 | **103.5** ±3.6 |
| walker2d-r | **87.7** ±14.1 | 74.3 ±13.9 | 79.2 ±10.0 |
| halfcheetah-m | 95.5 ±1.5 | **96.6** ±0.9 | **96.7** ±1.4 |
| hopper-m | 91.4 ±27.8 | 93.6 ±13.9 | **106.9** ±1.5 |
| walker2d-m | 121.6 ±2.3 | **124.1** ±2.4 | 122.8 ±3.0 |
| halfcheetah-m-r | 90.1 ±1.3 | 90.0 ±1.4 | **91.6** ±2.1 |
| hopper-m-r | 78.9 ±24.5 | 94.7 ±26.8 | **107.4** ±1.9 |
| walker2d-m-r | 119.0 ±2.2 | **122.5** ±2.7 | 120.9 ±2.3 |
| MuJoCo total | 866.0 | 887.6 | **918.7** |

demonstrated state-of-the-art performances through the use of ensemble techniques and a high UTD ratio. To assess the performance of integrating OPT with RLPD, we extend RLPD by incorporating an offline phase, followed by online fine-tuning, and subsequently apply OPT to evaluate its effectiveness. Further implementation details are provided in Appendix A.3.

Table 5 reports the results of original RLPD ('Vanilla'), RLPD with an additional offline phase ('Off-to-On'), and RLPD combined with OPT ('OPT'). The experimental results demonstrate that integrating OPT with RLPD leads to performance improvements, surpassing both the baseline methods. These results indicate that OPT is an effective algorithm capable of enhancing performance when applied to existing state-of-the-art algorithms.

## 6. Conclusion

In this work, we propose Online Pre-Training for Offline-to-Online RL (OPT), a novel approach designed to improve the fine-tuning of offline pre-trained agents. OPT introduces an additional learning phase, Online Pre-Training, which enables the training of a new value function prior to online fine-tuning. By leveraging this newly learned value function, OPT facilitates a more efficient fine-tuning process, effectively mitigating the performance degradation commonly observed in conventional offline-to-online RL approaches. Through extensive experiments across various D4RL environments, we demonstrate that OPT consistently outperforms existing methods and exhibits versatility across different backbone algorithms. These results highlight OPT as a robust and effective solution for enhancing performance in offline-to-online RL. Our findings further emphasize incorporating a new value function and leveraging it through Online Pre-Training to improve fine-tuning efficiency. We believe that OPT paves the way for future research on refining fine-tuning strategies in reinforcement learning.

## Acknowledgements

This work was supported by LG AI Research and partly by the National Research Foundation of Korea (NRF) grant funded by the Korea government (MSIT) (No. RS-2025-00557589, Generative Model Based Efficient Reinforcement Learning Algorithms for Multi-modal Expansion in Generalized Environments).

## Impact Statement

This paper presents work whose goal is to advance the field of Machine Learning. There are many potential societal consequences of our work, none which we feel must be specifically highlighted here.

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

# A. Implementation

## A.1. Implementation Details for OPT

We implement OPT based on the codebase of each backbone algorithm. TD3 and RLPD are based on their official implementation[1][2], SPOT, IQL are built upon the CORL (Tarasov et al. 2024) library [3]. The primary modification introduced in our method is the addition of the Online Pre-Training phase. The Online Pre-Training phase is implemented by modifying the meta-adaptation method provided in the OEMA code[4], tailored specifically for value function learning. Additionally, the balanced replay (Lee et al. 2022) is implemented using the authors' official implementation[5]. Aside from these changes, no other alterations are made to the original code. The complete training code is available at https://github.com/LGAI-Research/opt.

## A.2. Additional Implementation Details for IQL

The proposed method is applicable to various baseline algorithms. Here, we present the implementation when applied to IQL. IQL trains the state value function and the state-action value function as follows:

$$L_V(\mu) = \mathbb{E}_{(s,a)\sim\mathcal{D}}\left[\mathcal{L}_2^\tau(Q_{\bar\theta}(s,a) - V_\mu(s))\right] \tag{5}$$

$$L_Q(\theta) = \mathbb{E}_{(s,a,r,s')\sim\mathcal{D}}\left[(r + \gamma V_\mu(s') - Q_\theta(s,a))^2\right] \tag{6}$$

where $\mathcal{L}_2^\tau(u) = |\tau - 1(u < 0)|u^2$ and $\tau \in (0,1)$ is the expectile value, and $Q_{\bar\theta}$ denotes a target state-action value function. Then, using the state-action value function and state value function, the policy is trained through Advantage Weighted Regression:

$$L_\pi(\phi) = \mathbb{E}_{(s,a)\sim\mathcal{D}}\left[\exp(\beta(Q_{\bar\theta}(s,a) - V_\mu(s)))\log\pi_\phi(a|s)\right] \tag{7}$$

where $\beta \in [0,\infty)$ is an inverse temperature. To apply the proposed method to IQL, we train both a new state-action value function ($Q^{\text{on-pt}}$) and state value function ($V^{\text{on-pt}}$) in the Online Pre-Training. The state-action value function is trained identically to Eq. (3), while the state value function is trained as follows:

$$\mathcal{L}_V^{\text{pretrain}}(\mu) = \mathcal{L}_V^{\text{off}}(\mu) + \mathcal{L}_V^{\text{on}}(\mu - \alpha\nabla\mathcal{L}_V^{\text{off}}(\mu)). \tag{8}$$

In the online fine-tuning, policy improvement utilizes $\pi^{\text{off}}$, $Q^{\text{off-pt}}$, and $V^{\text{off-pt}}$ from offline pre-training as well as $Q^{\text{on-pt}}$ and $V^{\text{on-pt}}$ from Online Pre-Training. The policy is trained using an advantage weight obtained from $Q^{\text{off-pt}}$ and $V^{\text{off-pt}}$, and a separate weight obtained from $Q^{\text{on-pt}}$ and $V^{\text{on-pt}}$. These two sets of weights are then combined to effectively train the policy:

$$\begin{aligned}
L_\pi(\phi) = \mathbb{E}_{(s,a)\sim\mathcal{D}}\Big[\exp\big(\beta\big(\kappa(Q_{\bar\theta}^{\text{off-pt}}(s,a) - V_\mu^{\text{off-pt}}(s)) \\
+ (1-\kappa)(Q_{\bar\psi}^{\text{on-pt}}(s,a) - V_\nu^{\text{on-pt}}(s)))\big)\log\pi_\phi(a|s)\Big].
\end{aligned} \tag{9}$$

## A.3. Additional Implementation Details for RLPD

To verify the effectiveness of the proposed method when applied to ensemble techniques and high UTD ratio, we conduct experiments by integrating OPT into RLPD. Since RLPD does not originally include an offline phase, we incorporate an additional offline phase into its implementation. During the offline phase, we follow the RLPD learning method, performing 1M gradient steps. In the online phase, we adhere to OPT by conducting Online Pre-Training for 25k environment steps, followed by 275k environment steps. Given that the original RLPD employs symmetric sampling in the online phase, where half of the samples are sampled from offline data and the other half from online data, we also utilize symmetric sampling instead of balanced replay.

---

[1]https://github.com/sfujim/TD3

[2]https://github.com/ikostrikov/rlpd

[3]https://github.com/tinkoff-ai/CORL

[4]https://github.com/guosyjlu/OEMA

[5]https://github.com/shlee94/Off2OnRL

### A.4. Baseline Implementation

For comparison with OPT, we re-run all baselines. The results for all baselines are obtained using their official implementations.

- OFF2ON (Lee et al. 2022): `https://github.com/shlee94/Off2OnRL`

- OEMA (Guo et al. 2023): `https://github.com/guosyjlu/OEMA`

- PEX (Zhang et al. 2023): `https://github.com/Haichao-Zhang/PEX`

- ACA (Yu & Zhang 2023): `https://github.com/ZishunYu/Actor-Critic-Alignment`

- FamO2O (Wang et al. 2024): `https://github.com/LeapLabTHU/FamO2O`

- Cal-QL (Nakamoto et al. 2024): `https://github.com/nakamotoo/Cal-QL`

## B. Detailed Description of the Environment and Dataset.

### B.1. MuJoCo

MuJoCo consists of locomotion tasks and provides datasets of varying quality for each environment. We conduct experiments on the `halfcheetah`, `hopper`, and `walker2d` environments. MuJoCo environment are dense reward setting, and we use the "-v2" versions of the `random`, `medium`, and `medium-replay` datasets for each environment.

### B.2. Antmaze

Antmaze involves controlling an ant robot to navigate from the start of the maze to the goal. Antmaze is a sparse reward environment where the agent receives a reward of +1 upon reaching the goal, and 0 otherwise. The maze is composed of three environments: umaze, medium, and large. For the dataset, we use the "-v2" versions of `umaze`, `umaze-diverse`, `medium-play`, `medium-diverse`, `large-play`, and `large-diverse`.

### B.3. Adroit

Adroit is a set of tasks controlling a hand robot with five fingers. Each environment has a different objective: in the `pen` environment, the task is twirling a pen; in the `hammer` environment, hammering a nail; in the `door` environment, grabbing a door handle and opening it; and in the `relocate`, locating a ball to goal region. We utilize the "-v1" version of the `cloned` dataset for each environment.

## C. Further Experimental Results

### C.1. Ablation on Replay Buffer

The proposed method employs balanced replay (Lee et al. 2022) to enhance the use of online samples during online fine-tuning. Since the balanced replay has proven effective in offline-to-online RL on its own, we conduct experiments to assess its impact and dependency within OPT. To align with the original framework of OPT, which emphasizes rapid adaptation through more frequent learning for online samples during online fine-tuning, we test a setup that uniformly samples from online data, a method we refer to as Online Replay (OR).

*Table 6.* Comparing normalized score for OPT with balanced replay and online replay buffer. All results are reported as the mean and standard deviation across five random seeds. `Improvement(%)` refers to the performance gain when compared to the baseline.

| Environment | OPT with BR | OPT with OR |
|---|---|---|
| halfcheetah-r | **89.0** $_{\pm 2.1}$ | 81.4 $_{\pm 6.2}$ |
| hopper-r | 109.5 $_{\pm 3.1}$ | **109.5** $_{\pm 5.1}$ |
| walker2d-r | 88.1 $_{\pm 5.2}$ | **89.7** $_{\pm 21.4}$ |
| halfcheetah-m | **96.6** $_{\pm 1.7}$ | 89.8 $_{\pm 3.0}$ |
| hopper-m | 112.0 $_{\pm 1.3}$ | **111.2** $_{\pm 1.7}$ |
| walker2d-m | **116.1** $_{\pm 4.7}$ | 113.9 $_{\pm 8.2}$ |
| halfcheetah-m-r | **92.2** $_{\pm 1.2}$ | 85.3 $_{\pm 2.7}$ |
| hopper-m-r | **112.7** $_{\pm 1.1}$ | 111.4 $_{\pm 1.8}$ |
| walker2d-m-r | **117.7** $_{\pm 3.5}$ | 109.1 $_{\pm 9.1}$ |
| Total | **933.9** | 901.3 |
| Improvement (%) | **24.5** % | 20.2 % |

The results in Table 6 demonstrate that the performance gains of OPT are not solely attributable to the effects of balanced replay. Furthermore, replacing balanced replay with this simpler Online Replay setup still results in significant performance improvements compared with the baseline. These findings indicate that the performance of OPT stems not just from balanced replay, but from other strategy that emphasizes learning from online samples, such as Online Replay.

### C.2. Ablation on $\kappa$

OPT adjusts the weights of $Q^{\text{off-pt}}$ and $Q^{\text{on-pt}}$ during online fine-tuning through the coefficient $\kappa$. To assess the necessity of $\kappa$ scheduling, we first conduct experiments with fixed $\kappa$ values of 0, 0.5, and 1. In addition, to analyze sensitivity to the scheduling parameters, we evaluate two alternative schedules where $\kappa$ increases linearly from 0.1 to 0.8 and from 0.1 to 0.7.

*Table 7.* Ablation study results on $\kappa$ in the Antmaze domain. All values are reported as the mean and 95% confidence interval over 5 random seeds.

| Environment | $0.1 \rightarrow 0.9$ | $0.1 \rightarrow 0.8$ | $0.1 \rightarrow 0.7$ | 0 | 1 | 0.5 |
|---|---|---|---|---|---|---|
| umaze | 99.8$_{\pm 0.3}$ | 100.0$_{\pm 0.0}$ | 99.6$_{\pm 0.7}$ | 95.2$_{\pm 7.0}$ | 98.8$_{\pm 1.1}$ | 99.0$_{\pm 0.6}$ |
| umaze-diverse | 97.4$_{\pm 0.3}$ | 96.8$_{\pm 1.2}$ | 95.2$_{\pm 3.2}$ | 77.6$_{\pm 14.1}$ | 94.0$_{\pm 1.2}$ | 97.0$_{\pm 2.1}$ |
| medium-play | 98.2$_{\pm 1.1}$ | 97.6$_{\pm 1.4}$ | 98.2$_{\pm 1.1}$ | 94.4$_{\pm 1.8}$ | 94.6$_{\pm 1.8}$ | 96.0$_{\pm 1.8}$ |
| medium-diverse | 98.4$_{\pm 1.1}$ | 97.6$_{\pm 1.3}$ | 96.6$_{\pm 2.1}$ | 96.0$_{\pm 2.0}$ | 95.2$_{\pm 2.6}$ | 97.0$_{\pm 1.3}$ |
| large-play | 78.2$_{\pm 3.8}$ | 78.3$_{\pm 3.9}$ | 81.0$_{\pm 6.4}$ | 46.0$_{\pm 20.5}$ | 41.8$_{\pm 16.7}$ | 59.2$_{\pm 25.2}$ |
| large-diverse | 90.6$_{\pm 3.2}$ | 90.0$_{\pm 1.3}$ | 88.4$_{\pm 3.2}$ | 52.0$_{\pm 19.6}$ | 49.0$_{\pm 17.2}$ | 75.4$_{\pm 8.4}$ |

The experimental results lead to two key observations. First, the comparison with fixed $\kappa$ values highlights the necessity of using a schedule. Second, the performance differences among various scheduled values are marginal, indicating that the precise choice of the $\kappa$ value is not critical. Rather, the crucial factor is the presence of a gradual transition between the two value functions.

These findings validate both the importance of $\kappa$ scheduling and the robustness of our method to variations in the scheduling configuration.

## C.3. Full Results for Comparison of Different Initialization Methods

In Section 5.1, we investigate the effectiveness of the Online Pre-Training strategy. We present the complete results of this experiment, which demonstrate that Online Pre-Training facilitates more efficient learning of $Q^{\text{on-pt}}$ compared to other baselines. This improvement is particularly pronounced in the random dataset, where the distribution shift is more significant.

*Table 8.* Results of the ablation study on Online Pre-Training. All experimental results are measured after 300k steps of online fine-tuning, with 5 random seeds used for each experiment.

| Environment | OPT | Random Initialization | Pre-trained with $B_{\text{on}}$ |
|---|---|---|---|
| halfcheetah-r | $89.0_{\pm 2.1}$ | $76.7_{\pm 11.9}$ | $68.3_{\pm 8.4}$ |
| hopper-r | $109.5_{\pm 3.1}$ | $105.8_{\pm 7.8}$ | $105.5_{\pm 3.2}$ |
| walker2d-r | $88.1_{\pm 5.2}$ | $74.8_{\pm 10.5}$ | $73.4_{\pm 28.9}$ |
| halfcheetah-m | $96.6_{\pm 1.7}$ | $89.7_{\pm 4.6}$ | $95.7_{\pm 0.8}$ |
| hopper-m | $112.0_{\pm 1.3}$ | $111.3_{\pm 1.9}$ | $109.6_{\pm 2.2}$ |
| walker2d-m | $116.1_{\pm 4.7}$ | $101.0_{\pm 9.2}$ | $114.7_{\pm 2.2}$ |
| halfcheetah-m-r | $92.2_{\pm 1.2}$ | $84.3_{\pm 8.2}$ | $91.0_{\pm 1.9}$ |
| hopper-m-r | $112.7_{\pm 1.1}$ | $111.7_{\pm 1.4}$ | $112.2_{\pm 0.7}$ |
| walker2d-m-r | $117.7_{\pm 3.5}$ | $104.8_{\pm 5.2}$ | $111.6_{\pm 8.4}$ |
| Total | 933.9 | 843.1 | 882.0 |

We further extend this evaluation to the Antmaze domain. In addition to the baselines considered in Section 5.1, we introduce a simplified alternative to our method by removing the meta-adaptation component from Equation (3). Specifically, the objective is replaced by:

$$\mathcal{L}_{Q^{\text{on-pt}}}^{\text{pretrain}}(\psi) = \mathcal{L}_{Q^{\text{on-pt}}}^{\text{off}}(\psi) + \mathcal{L}_{Q^{\text{on-pt}}}^{\text{on}}(\psi). \tag{10}$$

*Table 9.* Results of the ablation study on $N_{\tau}$. All experimental results are measured after 300k steps of online fine-tuning, with 5 random seeds used for each experiment.

| | OPT | Random Initialization | Pre-trained with $B_{\text{on}}$ | Pre-trained with $B_{\text{off}}$ and $B_{\text{on}}$ |
|---|---|---|---|---|
| umaze | $99.8_{\pm 0.3}$ | $99.6_{\pm 0.7}$ | $98.8_{\pm 1.2}$ | $99.4_{\pm 0.7}$ |
| umaze-diverse | $97.4_{\pm 0.3}$ | $95.4_{\pm 3.5}$ | $95.4_{\pm 2.6}$ | $96.6_{\pm 2.2}$ |
| medium-play | $98.2_{\pm 1.1}$ | $92.4_{\pm 2.0}$ | $95.6_{\pm 1.1}$ | $96.6_{\pm 2.0}$ |
| medium-diverse | $98.4_{\pm 1.1}$ | $92.8_{\pm 3.8}$ | $95.4_{\pm 1.5}$ | $97.2_{\pm 1.7}$ |
| large-play | $78.2_{\pm 3.8}$ | $1.0_{\pm 1.7}$ | $0.0_{\pm 0.0}$ | $0.0_{\pm 0.0}$ |
| large-diverse | $90.6_{\pm 3.2}$ | $33.8_{\pm 29.7}$ | $54.0_{\pm 27.1}$ | $15.2_{\pm 14.9}$ |
| Total | 562.6 | 415.0 | 439.2 | 405.0 |

As shown in Table 9, the results are consistent with those observed in Figure 5. Notably, the additional variant (pre-trained with both $B_{\text{off}}$ and $B_{\text{on}}$) exhibits a sharp decline in performance, especially in more challenging environments such as Antmaze-large. This degradation is attributed to conflicting learning dynamics between the two loss terms, which the simplified method fails to reconcile.

In contrast, our meta-adaptation objective resolves this conflict and enables more stable and effective online fine-tuning. These results underscore the importance of the meta-adaptation mechanism under distribution shift.

## C.4. Full Results for Comparison of Different $N_\tau$

In Section 5.5, we examine how the performance varies with different values of $N_\tau$. Table 10 presents the full results corresponding to Figure 12. The results show that increasing $N_\tau$ generally improves performance across all environments, but the performance tends to plateau beyond 25k steps.

*Table 10.* Results of the ablation study on $N_\tau$. All experimental results are measured after 300k steps of online fine-tuning, with 5 random seeds used for each experiment.

| | 6250 | 12500 | 25000 | 50000 | 100000 |
|---|---|---|---|---|---|
| halfcheetah-r | $66.4_{\pm 9.2}$ | $79.9_{\pm 4.8}$ | $89.0_{\pm 2.1}$ | $93.6_{\pm 2.1}$ | $93.7_{\pm 2.8}$ |
| hopper-r | $105.8_{\pm 1.3}$ | $103.7_{\pm 4.7}$ | $109.5_{\pm 3.1}$ | $108.4_{\pm 5.2}$ | $109.2_{\pm 2.0}$ |
| walker2d-r | $75.9_{\pm 8.1}$ | $72.9_{\pm 9.5}$ | $88.1_{\pm 5.2}$ | $92.4_{\pm 8.2}$ | $93.9_{\pm 4.0}$ |
| halfcheetah-m | $98.6_{\pm 1.6}$ | $97.0_{\pm 2.5}$ | $96.6_{\pm 1.7}$ | $98.4_{\pm 3.1}$ | $97.3_{\pm 2.1}$ |
| hopper-m | $105.8_{\pm 2.5}$ | $108.7_{\pm 2.0}$ | $112.0_{\pm 1.3}$ | $110.2_{\pm 1.6}$ | $110.9_{\pm 2.0}$ |
| walker2d-m | $112.8_{\pm 7.0}$ | $115.3_{\pm 5.1}$ | $116.1_{\pm 4.7}$ | $119.5_{\pm 4.8}$ | $116.2_{\pm 1.5}$ |
| halfcheetah-m-r | $89.9_{\pm 3.4}$ | $90.4_{\pm 1.4}$ | $92.2_{\pm 1.2}$ | $92.0_{\pm 1.8}$ | $92.6_{\pm 2.8}$ |
| hopper-m-r | $94.9_{\pm 5.5}$ | $111.1_{\pm 0.7}$ | $112.7_{\pm 1.1}$ | $112.8_{\pm 2.2}$ | $111.6_{\pm 1.5}$ |
| walker2d-m-r | $111.2_{\pm 2.3}$ | $113.0_{\pm 1.8}$ | $117.7_{\pm 3.5}$ | $113.2_{\pm 2.7}$ | $115.4_{\pm 2.8}$ |
| Total | 861.3 | 892.0 | 933.9 | 940.5 | 940.8 |

## C.5. Full Results for Ablation of Addition of a New Value Function

In Section 5.3, we analyze the impact of introducing a new value function on performance. Table 11 provides the full results corresponding to Figure 7. The results indicate that the addition of the new value function contributes to performance gains in almost all environments.

*Table 11.* Results of the ablation study on the addition of a new value function. All experimental results are measured after 300k steps of online fine-tuning, with 5 random seeds used for each experiment.

| | OPT | w/o $Q^{\text{on-pt}}$ |
|---|---|---|
| halfcheetah-r | $89.0_{\pm 1.8}$ | $94.0_{\pm 3.6}$ |
| hopper-r | $109.5_{\pm 2.7}$ | $96.9_{\pm 4.9}$ |
| walker2d-r | $88.1_{\pm 4.5}$ | $9.3_{\pm 9.5}$ |
| halfcheetah-m | $96.6_{\pm 1.4}$ | $93.1_{\pm 1.9}$ |
| hopper-m | $112.0_{\pm 1.1}$ | $109.0_{\pm 2.0}$ |
| walker2d-m | $116.1_{\pm 4.1}$ | $110.7_{\pm 2.8}$ |
| halfcheetah-m-r | $92.2_{\pm 1.0}$ | $90.6_{\pm 1.5}$ |
| hopper-m-r | $112.7_{\pm 0.9}$ | $109.5_{\pm 1.1}$ |
| walker2d-m-r | $117.7_{\pm 3.0}$ | $107.5_{\pm 2.9}$ |
| Total | 933.9 | 820.6 |

# D. Additional Analysis

Since the motivation of our method is to improve value estimation during online fine-tuning, we conduct additional experiments to verify whether the combined value function benefits from OPT. To evaluate this, we compare the value estimation of TD3 and TD3+OPT in three environments where OPT shows significant performance gains: halfcheetah-medium-replay-v2, hopper-random-v2, and walker2d-medium-v2. As an approximation of the optimal value function, we train a TD3 agent with sufficient steps until it reaches near-optimal performance and use its value function as a reference. For a fair comparison, we sample 10 fixed state-action pairs, where states are drawn from the initial state distribution and actions are selected using the optimal policy. We then measure the value estimation bias of TD3 and TD3+OPT throughout online fine-tuning.

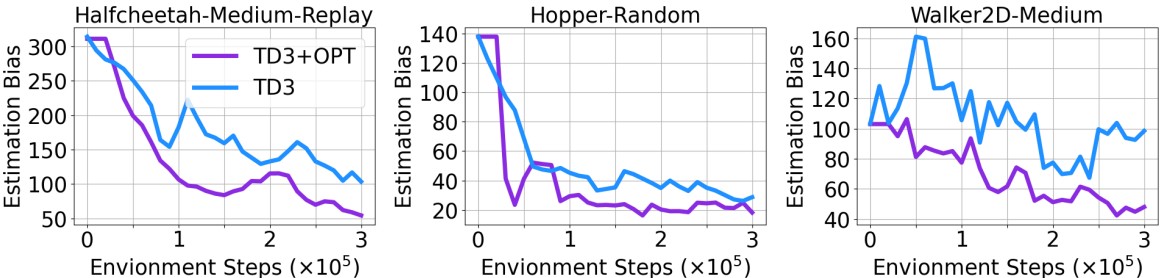

*Figure 9.* Estimation bias of value function, comparing TD3 and TD3+OPT against optimal value function. The bias for the OPT remains initially flat due to the Online Pre-Training phase.

Figure 9 presents how the estimation bias evolves during online fine-tuning. These results indicate that TD3 exhibits noticeable esimation bias due to distribution shift, whereas our method reduces bias early in training by leveraging $Q^{\text{on-pt}}$ trained specifically to handle online samples. To further support this, we conduct an additional experiment using the same estimation bias metric, but this time comparing $Q^{\text{off-pt}}$ and $Q^{\text{on-pt}}$ individually.

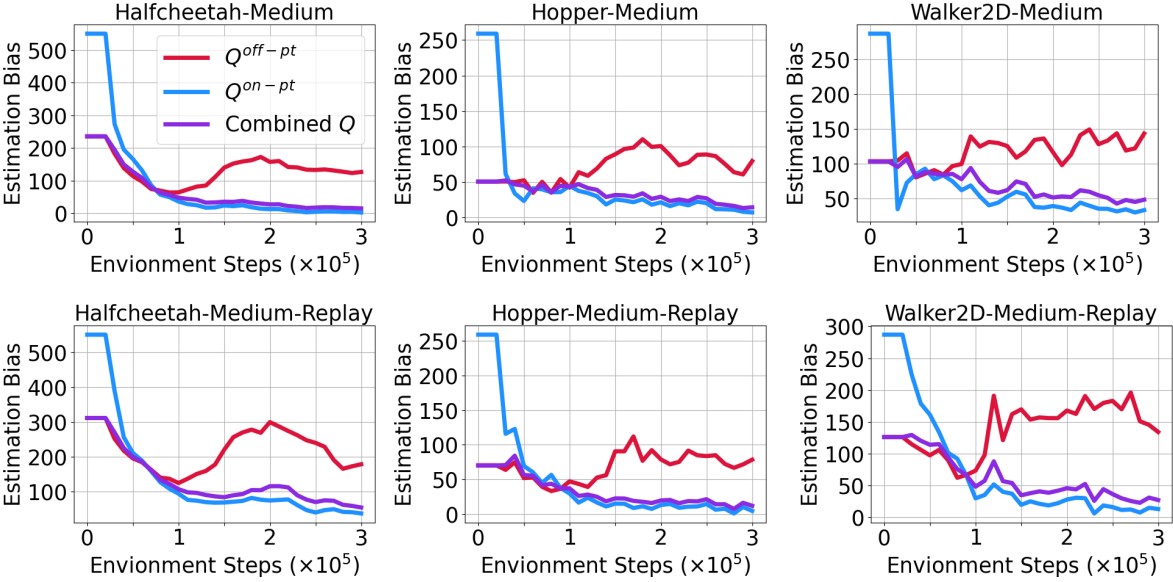

*Figure 10.* Estimation bias of the value function, comparing $Q^{\text{off-pt}}$, $Q^{\text{on-pt}}$, and the combined Q against the optimal value function. The combined Q corresponds to the formulation used in OPT, as defined in Equation (4) of the main paper.

As shown in Figure 10, $Q^{\text{on-pt}}$ reduces estimation bias more rapidly than $Q^{\text{off-pt}}$, indicating that the improvement observed above stems from the effective adaptation of $Q^{\text{on-pt}}$ to online samples. These findings confirm that our method enhances value estimation, leading to more stable and effective online fine-tuning.

## E. Comparison with Backbone Algorithm

Our proposed OPT is an algorithm applicable to value-based backbone algorithms. To evaluate the performance improvement achieved by applying OPT, we compare it against the performance of the backbone algorithms. Table 12 presents the performance of OPT when TD3, SPOT, and IQL are used as the backbone algorithms across all environments. When based on TD3 and SPOT, we observe an average performance improvement of 30%. Additionally, when based on IQL, we observe an average performance improvement of 25%.

*Table 12.* Comparison of normalized scores after online fine-tuning for each environment on the D4RL benchmark. We denote the backbone algorithm as `Vanilla` and the result of the algorithm integrated with OPT as `Ours`. All results are reported as the mean and 95% confidence interval over 10 random seeds.

| Environment | TD3 | | IQL | |
|---|---|---|---|---|
| | Vanilla | Ours | Vanilla | Ours |
| halfcheetah-r | **94.6** $\pm 2.7$ | 90.2 $\pm 1.6$ | 32.8 $\pm 1.6$ | **45.9** $\pm 2.5$ |
| hopper-r | 86.0 $\pm 13.3$ | **108.7** $\pm 1.7$ | 11.1 $\pm 1.1$ | **15.1** $\pm 3.8$ |
| walker2d-r | 0.1 $\pm 0.1$ | **88.0** $\pm 3.2$ | 6.6 $\pm 1.0$ | **9.8** $\pm 2.3$ |
| halfcheetah-m | 93.4 $\pm 1.9$ | **97.0** $\pm 0.8$ | 50.1 $\pm 0.1$ | **55.1** $\pm 0.4$ |
| hopper-m | 89.3 $\pm 14.5$ | **112.2** $\pm 0.6$ | 63.2 $\pm 3.5$ | **96.2** $\pm 5.2$ |
| walker2d-m | 103.5 $\pm 3.8$ | **116.1** $\pm 1.9$ | 83.9 $\pm 3.0$ | **90.4** $\pm 1.6$ |
| halfcheetah-m-r | 87.2 $\pm 0.9$ | **94.3** $\pm 1.5$ | 46.2 $\pm 0.2$ | **50.2** $\pm 1.1$ |
| hopper-m-r | 94.5 $\pm 11.3$ | **112.7** $\pm 0.4$ | **92.9** $\pm 8.2$ | 90.5 $\pm 11.2$ |
| walker2d-m-r | 103.9 $\pm 6.6$ | **119.9** $\pm 2.2$ | 91.9 $\pm 2.9$ | **105.3** $\pm 1.8$ |
| MuJoCo total | 752.4 | **939.1** (+24.8%) | 478.7 | **558.5** (+16.6%) |
| | SPOT | | IQL | |
| | Vanilla | Ours | Vanilla | Ours |
| umaze | 98.7 $\pm 0.8$ | **99.7** $\pm 0.2$ | 90.0 $\pm 2.7$ | **94.8** $\pm 1.7$ |
| umaze-diverse | 55.9 $\pm 15.9$ | **97.7** $\pm 0.4$ | 34.5 $\pm 11.5$ | **89.5** $\pm 7.2$ |
| medium-play | 91.1 $\pm 2.3$ | **97.6** $\pm 0.8$ | 84.2 $\pm 2.4$ | **89.5** $\pm 1.1$ |
| medium-diverse | 92.0 $\pm 1.7$ | **98.7** $\pm 0.6$ | 81.4 $\pm 2.4$ | **88.0** $\pm 1.7$ |
| large-play | 58.0 $\pm 11.4$ | **81.5** $\pm 4.5$ | 50.1 $\pm 5.3$ | **64.4** $\pm 1.9$ |
| large-diverse | 70.0 $\pm 11.5$ | **90.1** $\pm 2.9$ | 49.5 $\pm 3.5$ | **62.9** $\pm 3.7$ |
| Antmaze total | 465.7 | **565.3** (+21.3%) | 389.7 | **489.1** (+25.5%) |
| pen-cloned | 114.8 $\pm 6.3$ | **136.2** $\pm 4.8$ | 93.8 $\pm 7.5$ | **104.6** $\pm 5.9$ |
| hammer-cloned | 84.0 $\pm 13.8$ | **121.9** $\pm 1.6$ | 13.6 $\pm 3.8$ | **22.0** $\pm 11.9$ |
| door-cloned | 1.6 $\pm 2.7$ | **51.1** $\pm 12.9$ | 6.6 $\pm 2.2$ | **27.9** $\pm 4.8$ |
| relocate-cloned | -0.19 $\pm 0.02$ | **-0.06** $\pm 0.03$ | 0.06 $\pm 0.02$ | **0.77** $\pm 0.42$ |
| Adroit total | 200.2 | **309.1** (+54.4 %) | 117.1 | **155.3** (+32.6%) |

# F. Full Results of RLPD

In Section 5.6, we compare our proposed method with RLPD, which, despite being based on a different framework, has demonstrated impressive performance. Table 13 presents the full results of this comparison. The results further highlight that OPT can serve as a complementary component to enhance the effectiveness of established state-of-the-art methods.

*Table 13.* Average normalized final evaluation score for each environment on the D4RL benchmark. We denote the vanilla algorithm as `Vanilla`, the baseline algorithm within the offline-to-online RL framework as `Off-to-on`, and the result of the algorithm integrated with OPT as `Ours`. All results are reported as the mean and standard deviation across five random seeds.

| Environment | RLPD | | |
|---|---|---|---|
| | Vanilla | Off-to-on | Ours |
| halfcheetah-r | $91.5_{\pm 2.5}$ | $\mathbf{96.1}_{\pm 5.2}$ | $90.7_{\pm 2.2}$ |
| hopper-r | $90.2_{\pm 19.1}$ | $95.7_{\pm 18.4}$ | $\mathbf{103.5}_{\pm 3.6}$ |
| walker2d-r | $\mathbf{87.7}_{\pm 14.1}$ | $74.3_{\pm 13.9}$ | $79.2_{\pm 10.0}$ |
| halfcheetah-m | $95.5_{\pm 1.5}$ | $96.6_{\pm 0.9}$ | $\mathbf{96.7}_{\pm 1.4}$ |
| hopper-m | $91.4_{\pm 27.8}$ | $93.6_{\pm 13.9}$ | $\mathbf{106.9}_{\pm 1.5}$ |
| walker2d-m | $121.6_{\pm 2.3}$ | $\mathbf{124.1}_{\pm 2.4}$ | $122.8_{\pm 3.0}$ |
| halfcheetah-m-r | $90.1_{\pm 1.3}$ | $90.0_{\pm 1.4}$ | $\mathbf{91.6}_{\pm 2.1}$ |
| hopper-m-r | $78.9_{\pm 24.5}$ | $94.7_{\pm 26.8}$ | $\mathbf{107.4}_{\pm 1.9}$ |
| walker2d-m-r | $119.0_{\pm 2.2}$ | $\mathbf{122.5}_{\pm 2.7}$ | $120.9_{\pm 2.3}$ |
| MuJoCo total | 866.0 | 887.6 | **918.7** |
| umaze | $99.4_{\pm 0.8}$ | $\mathbf{99.8}_{\pm 0.4}$ | $99.6_{\pm 0.5}$ |
| umaze-diverse | $98.0_{\pm 1.1}$ | $\mathbf{99.2}_{\pm 1.0}$ | $99.0_{\pm 0.6}$ |
| medium-play | $97.6_{\pm 1.4}$ | $97.4_{\pm 1.4}$ | $\mathbf{99.6}_{\pm 0.6}$ |
| medium-diverse | $97.6_{\pm 1.9}$ | $98.6_{\pm 1.4}$ | $\mathbf{99.2}_{\pm 0.4}$ |
| large-play | $\mathbf{93.6}_{\pm 2.4}$ | $93.0_{\pm 2.5}$ | $92.2_{\pm 3.9}$ |
| large-diverse | $92.8_{\pm 3.2}$ | $90.4_{\pm 3.9}$ | $\mathbf{94.8}_{\pm 2.2}$ |
| Antmaze total | 579.0 | 578.4 | **584.4** |
| pen-cloned | $154.8_{\pm 11.6}$ | $148.5_{\pm 15.2}$ | $\mathbf{155.5}_{\pm 11.0}$ |
| hammer-cloned | $139.7_{\pm 5.6}$ | $141.4_{\pm 1.0}$ | $\mathbf{142.1}_{\pm 1.2}$ |
| door-cloned | $110.8_{\pm 6.1}$ | $114.6_{\pm 1.3}$ | $\mathbf{115.7}_{\pm 1.4}$ |
| relocate-cloned | $4.8_{\pm 7.1}$ | $0.11_{\pm 0.2}$ | $\mathbf{10.0}_{\pm 6.4}$ |
| Adroit total | 410.1 | 404.6 | **423.3** |

## G. Extra Comparison

This section presents a comparative analysis of OPT against recent baselines, including BOORL (Hu et al. 2024) and SO2 (Zhang et al. 2024). BOORL employs a Bayesian approach to address the trade-off between optimism and pessimism in offline-to-online reinforcement learning, providing a comprehensive framework for tackling this fundamental challenge. Similarly, SO2 incorporates advanced techniques into Q-function training to mitigate Q-value estimation errors, offering an effective strategy for improving value learning.

To ensure a fair comparison, experiments are conducted using the official implementations of BOORL and SO2, with hyperparameters set according to their original authors' recommendations. Additionally, the evaluation of these baselines are extended to 300k steps to align with the experimental setup of the proposed method. The results are summarized in Table 14.

*Table 14.* Comparison of the normalized scores after online fine-tuning for each environment in MuJoCo domain. All results are reported as the mean across five random seeds.

| | TD3+OPT 300k (ours) | BOORL 200k (paper) | BOORL 200k (reproduced) | BOORL 300k (reproduced) | SO2 100k (paper) | SO2 100k (reproduced) | SO2 300k (reproduced) |
|---|---|---|---|---|---|---|---|
| halfcheetah-r | 89.0 | 97.7 | 97.9 | 101.3 | 95.6 | 99.1 | 130.2 |
| hopper-r | 109.5 | 75.7 | 102.2 | 108.5 | 79.9 | 96.2 | 88.5 |
| walker2d-r | 88.1 | 93.6 | 4.3 | 0.1 | 62.9 | 36.9 | 66.8 |
| halfcheetah-m | 96.6 | 98.7 | 100.4 | 102.0 | 98.9 | 98.1 | 130.6 |
| hopper-m | 112.0 | 109.8 | 110.4 | 110.7 | 101.2 | 102.5 | 82.1 |
| walker2d-m | 116.1 | 107.7 | 105.3 | 114.5 | 107.6 | 109.1 | 105.7 |
| halfcheetah-m-r | 92.2 | 91.5 | 88.5 | 91.7 | 89.4 | 98.0 | 113.1 |
| hopper-m-r | 112.7 | 111.1 | 110.9 | 111.0 | 101.0 | 101.8 | 74.4 |
| walker2d-m-r | 117.7 | 114.4 | 109.6 | 112.3 | 98.2 | 96.8 | 27.1 |
| Total | 933.9 | 900.2 | 829.5 | 852.1 | 834.7 | 838.5 | 818.5 |

The results indicate that both BOORL and SO2 demonstrate competitive performance in specific environments, such as `halfcheetah`. Still, they generally underperform compared to OPT regarding average performance across all tasks. Notably, SO2 performs well at 100k steps; however, its effectiveness declines significantly when extended to 300k steps. BOORL achieves comparable performance to the proposed method in several environments but shows inconsistencies in reproducibility particularly in the `walker2d-random` task. Both BOORL and SO2 adopt ensemble-based approaches, resulting in significantly higher computational costs during training. In contrast, OPT achieves comparable or superior performance without relying on such ensemble techniques, making it a more efficient and effective solution.

# H. Hyper-parameters

In this paper, we present the results of applying OPT to various backbone algorithms. Aside from the hyperparameters listed below, all other hyperparameters are adopted directly from the backbone algorithms. In our proposed Online Pre-Training, we set $N_\tau$ to 25k and $N_{\text{pretrain}}$ to 50k for all environments. Additionally, for the MuJoCo domain, we use TD3 with a UTD ratio of 5 as the baseline. In the Adroit domain, we use SPOT as the baseline, trained with layer normalization (Ba 2016) applied to both the actor and critic networks. Except for these modifications, all other components and training details are kept identical to the original backbone algorithms.

As mentioned in Section 3.2, we use the parameter $\kappa$ to assign higher weight to $Q^{\text{off-pt}}$ during the early stages of online fine-tuning, gradually shifting to give higher weight to $Q^{\text{on-pt}}$ as training progresses. We control $\kappa$ through linear scheduling. Table 15 outlines the $\kappa$ scheduling for each environment, where $\kappa_{init}$ represents the initial value of $\kappa$ at the start of online phase, $T_{decay}$ specifies the number of timesteps over which $\kappa$ increases, and $\kappa_{end}$ indicates the final value of $\kappa$ after increase. Notably, in the MuJoCo random environment, as demonstrated in Section 5.2 the value function pre-trained offline exhibits significant bias, so it is not utilized during online fine-tuning.

*Table 15.* $\kappa$ scheduling method for each environment.

| Environment | $\kappa_{init}$ | $T_{decay}$ | $\kappa_{end}$ |
|---|---|---|---|
| halfcheetah-r | 1 | - | 1 |
| hopper-r | 1 | - | 1 |
| walker2d-r | 1 | - | 1 |
| halfcheetah-m | 0.3 | 150000 | 0.9 |
| hopper-m | 0.3 | 150000 | 0.9 |
| walker2d-m | 0.3 | 150000 | 0.9 |
| halfcheetah-m-r | 0.1 | 150000 | 0.9 |
| hopper-m-r | 0.1 | 150000 | 0.9 |
| walker2d-m-r | 0.1 | 150000 | 0.9 |
| umaze | 0.1 | 100000 | 0.9 |
| umaze-diverse | 0.1 | 100000 | 0.9 |
| medium-play | 0.1 | 100000 | 0.9 |
| medium-diverse | 0.1 | 100000 | 0.9 |
| large-play | 0.1 | 100000 | 0.9 |
| large-diverse | 0.1 | 200000 | 0.9 |
| pen-cloned | 0.1 | 250000 | 0.9 |
| hammer-cloned | 0.1 | 250000 | 0.9 |
| door-cloned | 0.1 | 250000 | 0.9 |
| relocate-cloned | 0.1 | 250000 | 0.9 |

# I. Learning Curves

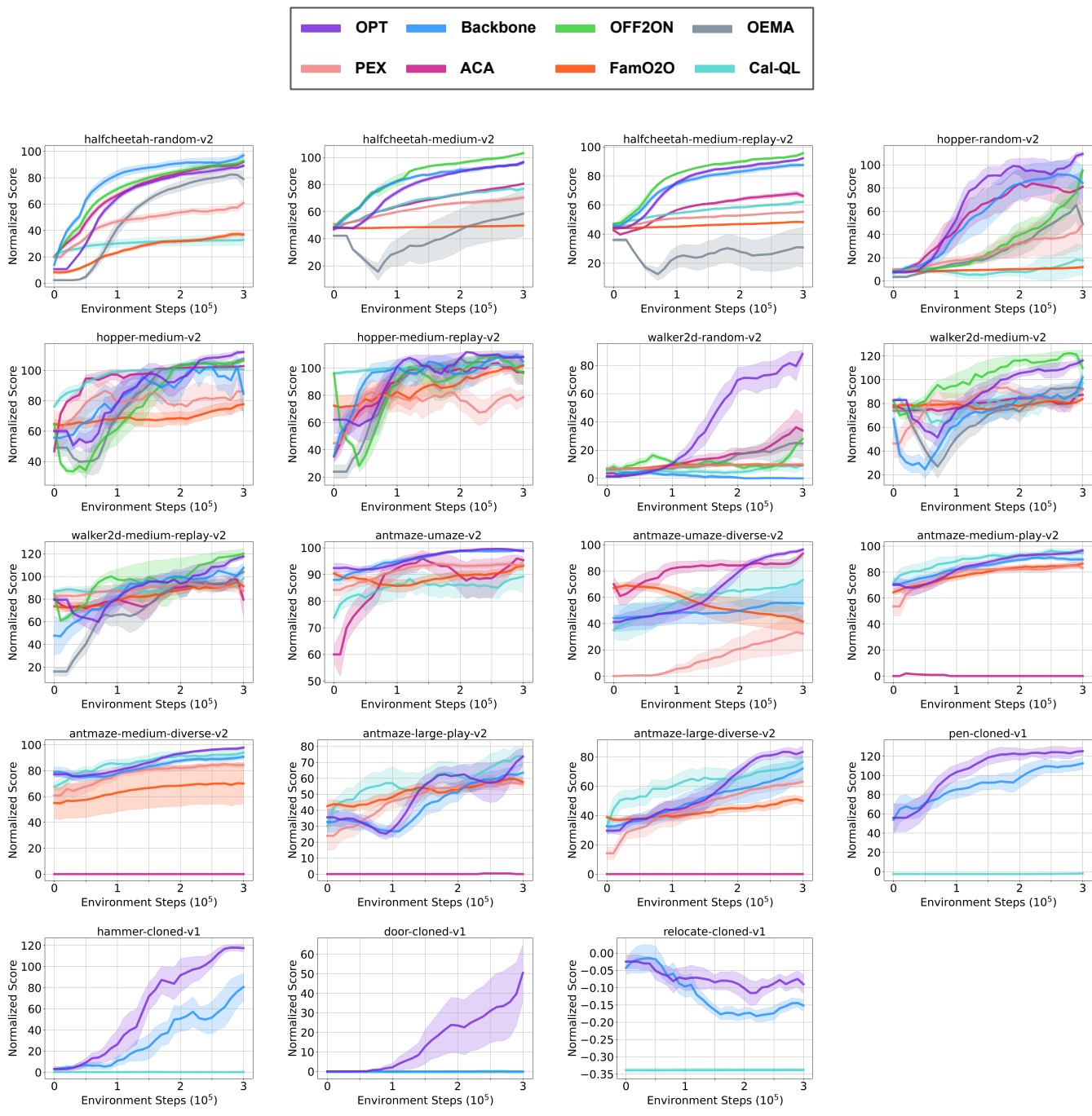

*Figure 11.* Learning curve of online phase over 300k steps. The solid line represents the mean performance, while the shaded region depicts the standard deviation across five random seeds.

## J. Computational Cost

To evaluate computational efficiency, we compare the wall-clock running time of TD3+OPT and TD3 in the walker2d-random-v2 environment, where TD3+OPT achieves the most significant performance gains.

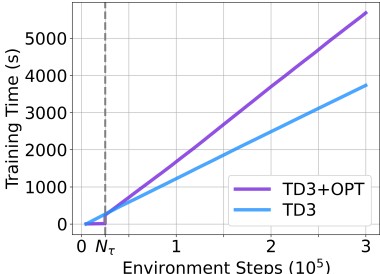

*Figure 12.* Comparison of wall-clock training time for TD3 and TD3 integrated with OPT on the walker2d-random-v2 environment using a single NVIDIA L40 GPU.

For TD3, the wall-clock time is approximately 4000 seconds. In contrast, TD3+OPT requires 6000 seconds, primarily due to the additional steps for Online Pre-Training ($N_\tau$ and $N_{\text{pretrain}}$ are set to 25k and 50k steps, respectively) and the updates associated with the addition of the new value function during online fine-tuning.

Although OPT requires additional time compared to TD3, this increase reflects the added effort to train the new value function, which is integral to enhancing adaptability and performance during online fine-tuning. The trade-off is balanced, as the additional computational cost results in substantial improvements in overall performance.

