# OpenReview forum: "Online Pre-Training for Offline-to-Online Reinforcement Learning"
_ICML.cc/2025/Conference — ICML 2025 poster_

### Official Review · Reviewer_2yWP · 2025-02-18

**Overall Recommendation:** 4

**Summary:**

The paper proposes a novel offline-to online RL approach where, at the end of the offline phase, a second critic is trained in an "Online pre-training" phase and used in addition to the offline dataset during the final online learning phase. During online pre-training, the offline policy and critic are frozen, and the new critic is trained from the frozen offline policy, filling a new online buffer. By leveraging both the offline critic and the pretrained online critic, offline-to online works better than with former approaches. The algorithm outperforms many state-of-the-art baselines in many environments.

## update after rebuttal:
I was in favor of accepting the paper, after reading all the reviews and exchanges with the authors, I'm still so.

**Claims And Evidence:**

The paper benefits from a strong methodology and the authors provide satisfactory evidence for most of their claims.

One exception that I found is in Figure 6 (ablation of the new value function) where the authors should provide information on the variability of the results and use an appropriate statistical test to check that there is a statistically significant difference between the full method and its ablation.

Another one is the study of the sensitivity of the algorithm to \kappa, which is far from clear (see comments below).

**Essential References Not Discussed:**

I could not find an essential missing reference

The authors should definitely read the WSRL paper:

Zhou, Z., Peng, A., Li, Q., Levine, S., & Kumar, A. (2024). Efficient online reinforcement learning fine-tuning need not retain offline data. arXiv preprint arXiv:2412.07762.

which has an alternative approach to the same problem, but they cannot be blamed for not doing so as this can be considered "concurrent work" (out less than 4 months ago).

**Experimental Designs Or Analyses:**

As stated above, I'm OK with all experimental design details

**Methods And Evaluation Criteria:**

Most experimental design details are satisfactory.

**Other Comments Or Suggestions:**

Clarity and writing concerns:
- 60% of the abstract says known things in the beginning, the authors could come quicker to the point of their paper.
- italic "counter-intuitive trends" is a poor short name for the phenomenon the authors focus on, as it is too general. There are many counter-intuitive trends in many domains, find something more specific to the context of this paper
- the "Background and related work" section is quite messy. This could be reorganized into two more clearly organized sections.
- Fig. 2 could probably be smaller, if the authors need more space in the main paper
- Section 4.1 should refer to Appendix H for the learning curves, close to the tables.
- If possible, Fig. 4 should move to p. 7.
- In Fig. 5, medium-replay and medium could be swapped, to be in the same order as the text mentions them. The figure is not much readable.
- There is an issue with explanations about \kappa defered to appendices C.2 and D: in Appendix C.2, the authors seem to compare a linear scheduling to another linear scheduling, the point they want to make is very unclear. And I believe all explanations about \kappa in Appendix D should move together with Appendix C.2.
- in 5.4 and appendices, all 25000 and 50000 should be rewritten 25K and 50K so that we see that it corresponds to the 25K online pre-training duration mentioned earlier


Typos and similar concerns:
- the authors use a lot the "'s" form, even for things that are not persons: "dataset's quality", "algorithm's code", etc.They shouldn't.
- the year is often missing in references, this seems to be done on purpose but I consider this as a bad practice.
- p. 5: ACA ... which pre-process ->processes
- 5.1 Comparison OPT... -> Comparing
- p. 7 "In particular, the results on the random dataset, where policy evolves more drastically as shown in Figure 5, the results demonstrate that an overfitted Qon-pt fails to learn with this policy improvement." -> fix this sentence!
- OPT is missing in the legend of Fig. 7
- caption Fig. 7 doamin -> domain
- A.1: TD3, RLPD -> TD3 and RLPD ... are based on its -> their
- not dots after an equation when the sentence is not finished ("... where...")
- use "\ eqref" rather than "\ ref" to refer to an equation.
- B.3 has a different objective(s)
- F: bayesian -> Bayesian

**Other Strengths And Weaknesses:**

Strengths:
- this is a solid paper, with a strong methodology (5 seeds, hyper-param studies, ablations, etc.) and a good contribution to the domain
- the method is quite simple though clever, and performs well
- beyond SOTA results, the additional studies in Section 5 are of interest


Weaknesses:
- writing could be improved here and there

**Questions For Authors:**

- Table 2, first row, umaze is umaze-play, right?
- In Figure 3, did the authors use a 300K + 25K + 275K protocol too? Are the first 300K steps shown? The transitions between offline, online pretraining and online should appear clearly, this is also the case of learning curves in Appendix H.

**Relation To Broader Scientific Literature:**

The paper is clearly positioned with respect to the relevant literature.

**Theoretical Claims:**

Does not apply (no theorem in this paper)

---

> ### Author Rebuttal · Authors · 2025-04-01
>
> We appreciate the reviewer for the valuable and constructive feedback.
> We respond to each point in detail below and will reflect the suggested improvements in the revised manuscript.
>
> &nbsp;
> ### **R4-1. Questions**
> - Question about umaze dataset
>
> According to the official D4RL[4-1] paper, the Antmaze environment consists of three types of datasets: umaze, diverse, and play.
> Among them, the ”play” dataset is collected by commanding the ant to navigate from various hand-picked start locations to hand-picked goals.
>
> In contrast, the *umaze* dataset, which appears in the first row of Table 2, uses a single fixed start and goal location without diversity in either.
> Since this setup does not match the definition of a *play* dataset, it is not categorized as one.
>
> We will clarify this point in the revised manuscript in Appendix B.2.
>
> - Question about training protocol
>
> All experiments in our paper follow a training protocol consisting of a 1M offline phase and a 300k online phase.
> For OPT, the first 25k steps of the online phase are used for online pre-training, while the remaining steps are allocated for online fine-tuning.
> This setup is also applied in Figure 3.
>
> Since the key changes in OPT occur during the online phase, Figure 3 and Appendix H present performance curves for the online phase.
> This is reflected in the initially flat performance observed in the graphs, as OPT does not update the policy during online pre-training.
>
> To avoid any confusion, we will explicitly clarify this in the caption of Figure 3 and in Appendix H.
>
> &nbsp;
> ### **R4-2. Sensitivity to $\kappa$**
> To provide clearer evidence for the sensitivity of our method, we conduct additional experiments in the Antmaze domain.
> Table F at the anonymous link (https://sites.google.com/view/icml2025opt/) highlights two key findings:
> - First, the comparison with fixed $\kappa$ values demonstrates the necessity of scheduling $\kappa$ during training.
> - Second, the results indicate that performance is not sensitive on a specific $\kappa$ value; rather, the crucial factor is the gradual transition from $Q^{\text{off-pt}}$ to $Q^{\text{on-pt}}$
>
> These findings validate both the importance of $\kappa$ scheduling and the robustness of the approach to sensitivity in its exact values.
>
> &nbsp;
> ### **R4-3. Regarding Writing and Reference**
> We sincerely thank the reviewer for the detailed and thoughtful feedback on the writing.
> We appreciate the time and care taken to point out areas for improvement, as well as for providing helpful references.
> All suggestions will be carefully considered and reflected in the revised manuscripts, and we believe these revisions will improve the clarity and overall quality of the paper.
>
> &nbsp;
> ### **R4-4. Regarding Figure 6**
> We thank the reviewer for pointing out the insufficient information provided in Figure 6 regarding variability and statistical significance.
>
> To provide further clarity, we include environment-specific results corresponding to Figure 6 in Table G in the anonymous link (https://sites.google.com/view/icml2025opt/).
> These results show that performance degradation is especially noticeable in the random dataset.
> We will include these additional results in the revised manuscript.
>
> &nbsp;
>
> Once again, we greatly appreciate the reviewer's thoughtful comments and suggestions. The feedback has been instrumental in improving the quality and clarity of our work. We hope our responses sufficiently address the concerns raised.
>
> &nbsp;
>
> [4-1] Fu, Justin, et al. "D4rl: Datasets for deep data-driven reinforcement learning." arXiv (2020).

---

### Official Review · Reviewer_Xjkr · 2025-03-13

**Overall Recommendation:** 3

**Summary:**

This paper presents OPT, a method to improve value estimation in RL. OPT follows three phases: offline pre-training, an "online pretraining" phase to train a separate value function, and online fine-tuning that combines both value functions. Unlike traditional methods that use a single value function, OPT introduces a second one trained with offline data and early online samples for better performance.

**Claims And Evidence:**

Several claims lack sufficient statistical significance:

1. The claim of "average 30% improvement" seems dubious when considering standard deviations, which often overlap significantly with baselines.
2. Claims of superiority over Cal-QL (lines 308-314) aren't statistically justified given overlapping standard deviations in Table 2.
3. Performance comparisons in Tables 1-3 highlight mean values without proper statistical significance testing, making it difficult to assess the consistency of improvements (this also includes Table 11 where authors claim superiority over BOORL without even reporting stdev).
4. The conclusion that OPT works well across backbone algorithms is supported by empirical results, but the improvements for IQL (Table 10) show substantial overlaps in standard deviations, weakening this claim.

**Essential References Not Discussed:**

-

**Experimental Designs Or Analyses:**

As I wrote in the "claims" section above, several issues with experimental design and analysis undermine the paper's conclusions:

1. Statistical reporting is inconsistent – standard deviations are missing entirely from Table 11 and Figure 4,6, making it impossible to assess the significance of comparisons.
2. The paper lacks an ablation study on whether Equation 4 (weighted combination of Q-functions) is necessary, given that κ scheduling was shown to have minimal impact (like on halfcheetah env).

Overall, I feel that there is a lot of overlap between the proposed method and other baselines if we factor in standard deviations, suggesting the improvements may not be statistically significant. And the selective reporting, i.e. highlighting only the mean values without statistical tests can create a misleading impression of consistent superiority. Therefore, empirically this paper does not seem sound to me.

**Methods And Evaluation Criteria:**

- The evaluation uses appropriate benchmarks (Mujoco, D4RL benchmarks) that are standard in the field.
- However, the meta-learning objective from OEMA appears unnecessarily complex without clear justification for why simpler approaches wouldn't work. The paper lacks ablations comparing this meta-learning approach to simpler alternatives.

**Other Comments Or Suggestions:**

Typos: line 165: "as" instead of "at"; line 167: "B_off." instead of "B_off,")

**Other Strengths And Weaknesses:**

## Strengths:

1. Novel approach to offline-to-online RL that addresses a fundamental limitation (inaccurate value estimation)
2. Extensive evaluation across multiple domains (MuJoCo, Antmaze, Adroit)
3. Good ablation studies on components like initialization methods and sample sizes
4. Paper is easy to follow

## Weaknesses:

1. Lack of theoretical justification for using two separate value functions (minor)
2. Insufficient comparison with simpler alternatives to the proposed complex approach (i.e. why do we need to use OEMA in equation 3?)
3. Selective reporting of results that emphasizes means over statistical significance and significant overlap with baselines

**Questions For Authors:**

1. How much additional computational cost is incurred when using OPT vs a baseline like TD3 or IQL?
2. Could you provide results using the rliable library [1]? The analysis can be made stronger with aggregated statistics like IQM rather than simply using means+stdev.
3. Why is the meta-adaptation objective from OEMA necessary? Have you compared with simpler approaches for training the new value function?
4. Can you provide "aggregated" learning curves with standard deviations for all baseline methods in Tables 1 and 2, similar to Figure 3, to allow for fair assessment of performance differences? (I was hoping for an aggregate curve in Appendix but could only find per-environment curves in Figure 8)

[1]: Agarwal, R., Schwarzer, M., Castro, P. S., Courville, A. C., & Bellemare, M. (2021). Deep reinforcement learning at the edge of the statistical precipice. Advances in neural information processing systems, 34, 29304-29320.

**Relation To Broader Scientific Literature:**

The problem statement is relevant to broader literature.

**Theoretical Claims:**

n/a

---

> ### Author Rebuttal · Authors · 2025-04-01
>
> We thank the reviewer for the constructive feedback, which helps improve both the clarity and completeness of our analysis.
> We address each point below.
>
> &nbsp;
> ### **R3-1. Regarding Statistical Significance**
> To reduce statistical uncertainty, we increase the number of random seeds from 5 to 10 for all baselines for Tables 1-3 and 10, and report 95% confidence intervals (CI) instead of standard deviation to better reflect the reliability of the results.
> Tables A-C at the anonymous link (https://sites.google.com/view/icml2025opt/) show that OPT consistently outperforms baselines, with minimal CI overlap, indicating statistical significance.
>
> Table E further supports this trend when combined with IQL, demonstrating robustness across backbone algorithms.
> For clarity, we bold entries whose CIs include the highest mean value among compared methods.
> We will include these results in the revised manuscript.
>
> &nbsp;
> ### **R3-2. Regarding Aggregated Statistics**
> We appreciate the suggestion to use aggregated metrics such as the Interquartile Mean (IQM) and include corresponding comparisons.
> Figure D at the anonymous link (https://sites.google.com/view/icml2025opt/) shows our method consistently achieves the strongest performance.
>
> We also provide aggregated learning curves.
> As shown in Figure E, our method demonstrates consistent improvement, especially strong result in the Adroit domain.
>
> These results support the statistical significance and overall effectiveness of our approach.
> We will include them in the revised manuscript.
>
> &nbsp;
> ### **R3-3. OPT with Simple Alternatives**
> As described in Section 3.1, Online Pre-Training initializes the new value function to enable effective online fine-tuning.
> To achieve this, we proposed a meta-learning strategy, formalized in Equation 3.
>
> The second term in Equation 3 updates the value function using online samples while incorporating gradients from $\mathcal{L}^{\text{off}}$.
> This aligns two terms, allowing $Q^{\text{on-pt}}$ to generalize across offline and online data.
> To assess its usefulness, we compare it with a simpler alternative that jointly trains on $B_{\text{off}}$ and $B_{\text{on}}$, as follows:
>
> $\mathcal{L}_{Q^{\text{on-pt}}}^{\text{pretrain}}= \mathcal{L}^{\text{off}}(\psi) + \mathcal{L}^{\text{on}}(\psi)$
>
> This experiment is shown in Table D of our anonymous link (https://sites.google.com/view/icml2025opt/).
> The variant **Pre-trained with $B_{\text{off}}$ and $B_{\text{on}}$** shows a performance drop in harder tasks e.g. large mazes,  due to conflicting learning dynamics between two terms, which the simpler method cannot resolve.
> In contrast, our method reconciles the two objectives, enabling effective online fine-tuning.
>
> These results highlight the value of the meta-adapation objective under distribution shift.
> We consider this a key contribution and will include this discussion in the revised manuscript.
>
> &nbsp;
> ### **R3-4. Ablation Study for Equation 4**
> We conducted experiments in Antmaze to assess the necessity of Eq. 4.
>
> Table F at the anonymous link (https://sites.google.com/view/icml2025opt/) compares $\kappa$ scheduling with fixed alternatives.
> While fixed $\kappa$ works in simple tasks, performance degrades in harder environments.
> In particular, $\kappa=0.5$ causes instability due to prolonged reliance on $Q^{\text{off-pt}}$.
>
> We also observe that performance is not sensitive to a specific $\kappa$ value, suggesting the key factor is a gradual shift from $Q^{\text{off-pt}}$ to $Q^{\text{on-pt}}$.
> These results underscore the importance of Equation 4 for stable and effective training.
>
> &nbsp;
> ### **R3-5. Computational Cost**
> We compare the wall-clock time of TD3 and our method.
> As shown in Figure F at the anonymous link (https://sites.google.com/view/icml2025opt/), TD3 takes about 4000 seconds, while ours takes around 6000 seconds due to the added value function and Online Pre-Training ($N_\tau = 25k$, $N_{\text{pretrain}}=50k$).
>
> Although this introduces extra overhead, we believe the performance gains reasonably justify the additional cost.
> We will include this analysis in the appendix.
>
> &nbsp;
> ### **R3-6. Regarding Theoretical Justification**
> Our work aims to empirically address challenges in offline-to-online RL.
> Although it does not include theoretical justification, the results support the effectiveness of our design.
> We recognize the value of theoretical analysis and consider it as a promising future direction.
>
> &nbsp;
> ### **R3-7. Regarding Missing Standard Deviation**
> We apologize for omitting standard deviations in Table 11 and Figures 4 and 6.
> We will include them in the revised manuscript for proper assessment.
>
> &nbsp;
>
> We appreciate the reviewer again for the valuable feedback and hope our response sufficiently addresses the concerns. We believe our method achieves statistically significant performance and is readily applicable to a wide range of algorithms. We hope this to be considered in the reviewer’s reevaluation of our work.

---

> > ### Comment · Reviewer_Xjkr · 2025-04-04
> >
> > Thank you for answering my concerns and reporting the scores using 95% CIs. Looking at Table A and Figure E in the anonymous link shared, it is evident that the method does not provide huge benefits over other baselines in Mujoco, but does help in Adroit and Antmaze (slightly). Therefore, I am raising my score to 3: Weak Accept.

---

> > > ### Author Response · Authors · 2025-04-05
> > >
> > > Thank you for your thoughtful follow-up and for reconsidering the score.
> > > We sincerely appreciate your careful evaluation based on the additional results, as well as your acknowledgment of the method’s improvements in Adroit and Antmaze.
> > > Your feedback has been very helpful in improving our work, and we are grateful for your time and constructive comments.

---

### Official Review · Reviewer_pJZg · 2025-03-14

**Overall Recommendation:** 4

**Summary:**

The paper studies offline-to-online RL problem and propose a new method called online pretraining. The mains to solve the sub-optimality problem brought by the offline Q value function during online fine-tuning. Specifically, the method propose to freeze the offline policy at the beginning of the online fine-tuning stage, collect online data and initialize the online Q value function with the combination of online and offline data with the OMEGA algorithm. During the rest of the online stage, the proposed method trains the policy with the combination of both the online Q value function and offline Q value function. The paper performs extensive empirical study on the effectiveness of  the proposed algorithm.

**Claims And Evidence:**

The paper makes claims on the strong performance of the proposed algorithm and it is validated by experiments on the relevant benchmarks over dense and sparse reward environments, and compared with a comprehensive set of baselines. The only point that might seem unfair is that the proposed method used different backbone algorithms for different tasks.

As for the effectiveness of each component of the proposed method, the paper also performs good ablation experiments on different initialization of the online Q functions. However, it would also be interesting to see if different initialization affects the asymptotic performance of the algorithm.

The decaying coefficient $\kappa$ seems like a very intuitive idea and the paper attempts to provide an analysis on this design choice as well. However, I do not believe that the current evidence makes sense because the offline Q function is also trained on the online data as well. Maybe one way is to show the difference between online/offline Q function to the optimal Q function along the training trajectory?

**Essential References Not Discussed:**

n/a

**Experimental Designs Or Analyses:**

yes

**Methods And Evaluation Criteria:**

As mentioned above, the paper already contains a very comprehensive set of evaluations. However, it will benefit the paper even further if the analysis experiments in Section 5 are also repeated on the whole set of environments.

**Other Comments Or Suggestions:**

According to Table 4, it seems like that RLPD is actually the strongest baseline in general (by a large margin)? Thus it would be nice if it is included in the comparison  in Table 1.

**Other Strengths And Weaknesses:**

n/a

**Questions For Authors:**

n/a

**Relation To Broader Scientific Literature:**

It provides a flexible method for offline2online RL with strong performance, which benefits the community.

**Theoretical Claims:**

n/a

---

> ### Author Rebuttal · Authors · 2025-04-01
>
> We appreciate the reviewer for the detailed and insightful comments.
> The suggestions are highly valuable in helping us refine both the presentation and the depth of our analysis.
> We carefully address each point below.
>
> &nbsp;
> ### **R2-1. Regarding Asymptotic Performance of Figure 4**
> We appreciate the reviewer for raising this interesting point.
> To investigate the asymptotic performance for different initialization strategies, we provide learning curves in Figure B at the anonymous link (https://sites.google.com/view/icml2025opt/).
> As discussed in Section 5.1 of our paper, **Random Initialization** initially causes an early performance drop due to unstable value estimates, which in turn results in lower asymptotic performance.
> **Pre-trained with $B_{\text{on}}$** achieves performance comparable to OPT in medium and medium-replay datasets but shows weak asymptotic performance on the random dataset.
> These results further support the effectiveness of the initialization strategy employed in our method.
>
> &nbsp;
> ### **R2-2. Regarding of $\kappa$ Scheduling**
> The motivation behind the design is that during online fine-tuning, $Q^{\text{off-pt}}$ becomes less reliable due to distribution shift, resulting in inaccurate value estimation.
> To address this, we introduce $\kappa$ scheduling strategy that gradually shifts emphasis toward $Q^{\text{on-pt}}$, enabling more effective online fine-tuning.
>
> We empirically validate this approach through an experiment comparing $Q^{\text{off-pt}}$ and $Q^{\text{on-pt}}$ against an optimal value function during the online fine-tuning.
> The optimal value function is estimated by training a TD3 agent until it achieves near-optimal performance.
> For a fair evaluation, we use 10 fixed state-action pairs, where states are sampled from the initial state distribution, and actions are selected using the optimal policy.
>
> As shown in Figure C at the anonymous link (https://sites.google.com/view/icml2025opt/), $Q^{\text{off-pt}}$ exhibits higher estimation bias due to its inability to adapt under distribution shift.
> In contrast, $Q^{\text{on-pt}}$, aided by Online Pre-Training, effectively reduces estimation bias.
> These results highlight that a gradual shift from $Q^{\text{off-pt}}$ to $Q^{\text{on-pt}}$ serves as an effective mechanism for mitigating estimation bias during online fine-tuning.
>
> &nbsp;
> ### **R2-3. Analysis Experiments on Other Domains**
> We agree that extending the analysis experiments in Section 5 to all environments can improve the comprehensiveness of our study.
> We are currently conducting analysis experiments, and one preliminary result is shown in Table D at the anonymous link (https://sites.google.com/view/icml2025opt/), which exhibits patterns consistent with those in Figure 4 of the main paper.
> We plan to include results covering all environments and analyses in the revised manuscript.
>
> &nbsp;
> ### **R2-4. Regarding Different Backbone Algorithms**
> The reason for using different backbone algorithms is that no single algorithm consistently performs well across all domains.
> For example, Off2On[2-1] performs well in MuJoCo but is not evaluated on Antmaze, and our reproduction did not perform well there.
> Conversely, SPOT[2-2] performs well in Antmaze but underperforms in MuJoCo.
> As each algorithm tends to be specialized for certain benchmarks, we choose the most suitable backbone for each to better assess the effect of our method.
>
> This is possible because our method is designed to be flexible and readily applicable to a wide range of algorithms.
> We will clarify this in the revised manuscript to avoid potential confusion.
>
> &nbsp;
> ### **R2-5. Regarding RLPD**
> Our work focuses on methods in the offline-to-online RL framework, which includes a distinct offline phase.
> For this reason, the current manuscript does not include RLPD[2-3] in the main results, as it is primarily designed for online RL settings and does not involve a separate offline phase.
> However, to prevent confusion in baseline comparisons, we will include RLPD in Table 1 in the revised manuscript, along with a brief explanation clarifying its distinction.
>
> &nbsp;
>
> We sincerely thank the reviewer again for the constructive comments. The feedback helped us strengthen both the empirical evidence and the clarity of our manuscript. We hope our responses have sufficiently addressed the concerns and contributed to a clearer understanding of our method.
>
> &nbsp;
>
> [2-1] Lee, Seunghyun, et al. "Offline-to-online reinforcement learning via balanced replay and pessimistic q-ensemble." CORL 2022.
>
> [2-2] Wu, Jialong, et al. "Supported policy optimization for offline reinforcement learning." NeurIPS 2024.
>
> [2-3] Ball, Philip J., et al. "Efficient online reinforcement learning with offline data." ICML 2023.

---

> > ### Comment · Reviewer_pJZg · 2025-04-04
> >
> > I appreciate the author's efforts in the rebuttal and I think the new results and analysis further improves the quality of the paper. I will raise my score accordingly.

---

> > > ### Author Response · Authors · 2025-04-04
> > >
> > > Thank you very much for your positve feedback and raising the score.  We appreciate the reviewer's time and effort for carefully reviewing our paper and response. Thank you again.

---

### Official Review · Reviewer_VzZL · 2025-03-18

**Overall Recommendation:** 3

**Summary:**

The authors proposed a new offline-to-online RL method called Online Pre-Training (OPT), where a new phase, online pertaining,  is added between offline pre-training and online fine-tuning to solve the inaccurate value estimation problem. OPT introduce a separate value function instead of directly continue the learning on the value function trained on offline dataset. Experiments are conducted on D4RL to confirm its superiority over other offline-to-online methods.

**Claims And Evidence:**

See questions.

**Essential References Not Discussed:**

N/A

**Experimental Designs Or Analyses:**

See questions.

**Methods And Evaluation Criteria:**

See questions.

**Other Comments Or Suggestions:**

N/A

**Other Strengths And Weaknesses:**

See questions.

**Questions For Authors:**

1. The experiments results for TD3 in table 1 , Vanilla RLPD in table 4 and other initializations in table 7 seem to come with significant larger standard deviation than OPT, taking the standard deviation into consideration, OPT seems not convincingly outperform the baselines on a lot of the tasks. Would the authors consider increase the number of random seeds?
2. The authors conducted ablation study for the weighting coefficient K on different dataset. However, for a new task, B_init and B_final are not available until you run the fine-tuning, making it hard to choose k before analysis. It would be great if authors could give a more intuitive guide of balancing strategies according to the task complexity of other properties.
3. The main motivation that authors propose the extra online pre-training stage with separate online Q function is that there are previous empirical experiments in papers like ((Nakamoto et al. 2024) showing the smoothness of Q function learning when switching from offline to online thus the algorithms suffer from an initial unlearning effect. I think it’s would be a necessary extra experiments to show the combined Q as in (3) used in online fine-tuning indeed improved after using OPT.

**Relation To Broader Scientific Literature:**

The idea of two separate Q function is innovative.

**Theoretical Claims:**

See questions.

---

> ### Author Rebuttal · Authors · 2025-04-01
>
> We sincerely thank the reviewer for the thoughtful and constructive feedback.
> The comments help improve the clarity and completeness of our work.
> Below, we address each point in detail.
>
> &nbsp;
> ### **R1-1. Regarding Experimental Results**
> Following the reviewer’s suggestion, we increase the number of random seeds from 5 to 10 to strengthen the claim regarding the effectiveness of OPT.
>
> In Table A-C of the anonymous link (https://sites.google.com/view/icml2025opt/), we report the mean and 95% confidence interval (CI) rather than standard deviation to convey the reliability of the results.
> OPT consistently outperforms the baseline, and the minimal CI overlap supports the statistical significance of the results.
>
> We also provide Interquartile Mean (IQM) [1-1] comparisons in Figure D for each domain, which further confirm that our method achieves strong performance.
> Additionally, we will increase the number of random seeds for Tables 4 and 7 and include the corresponding updated results in the revised manuscript.
>
> &nbsp;
> ### **R1-2. Intuitive guide of balancing strategies**
> We appreciate the reviewer for pointing out this insightful consideration.
> Although we do not provide a theoretical analysis for selecting the weighting coefficient $\kappa$, we offer the following intuitive guideline based on our empirical observations:
>
> - If the dataset quality is moderate, we recommend setting $\kappa$ such that the emphasis on $Q^{\text{on-pt}}$ gradually increases during online fine-tuning.
> - Conversely, if the dataset quality is relatively low, it is beneficial to set $\kappa$ to utilize $Q^{\text{on-pt}}$ from the early stage of online fine-tuning.
>
> We will include this practical guideline in the revised manuscript to provide clearer guidance on applying our method across various tasks.
>
> &nbsp;
> ### **R1-3. Regarding Additional Experiments on Value Estimation**
> Since the motivation of our method is to improve value estimation during online fine-tuning, we conduct additional experiments to verify whether the combined value function benefits from OPT.
>
> To evaluate this, we compare the value estimation of TD3 and TD3+OPT in three environments where OPT shows significant performance gains: halfcheetah-medium-replay-v2, hopper-random-v2, and walker2d-medium-v2.
> As an approximation of the optimal value function, we train a TD3 agent with sufficient steps until it reaches near-optimal performance and use its value function as a reference.
>
> For a fair comparison, we sample 10 fixed state-action pairs, where states are drawn from the initial state distribution and actions are selected using the optimal policy.
> We then measure the value estimation bias of TD3 and TD3+OPT throughout online fine-tuning.
>
> Figure A of the anonymous link (https://sites.google.com/view/icml2025opt/) presents how the estimation bias evolves during online fine-tuning.
> These results indicate that TD3 exhibits noticeable esimation bias due to distribution shift, whereas our method reduces bias early in training by leveraging $Q^{\text{on-pt}}$ trained specifically to handle online samples.
>
> To further support this, we conduct an additional experiment using the same estimation bias metric, but this time comparing $Q^{\text{off-pt}}$ and $Q^{\text{on-pt}}$ individually.
> As shown in Figure C, $Q^{\text{on-pt}}$ reduces estimation bias more rapidly than $Q^{\text{off-pt}}$, indicating that the improvement observed above stems from the effective adaptation of $Q^{\text{on-pt}}$ to online samples.
>
> These findings confirm that our method enhances value estimation, leading to more stable and effective online fine-tuning.
> We will include this analysis in the revised manuscript to support our claims.
>
> &nbsp;
>
> We appreciate the reviewer's detailed and insightful feedback once again. The suggestions have meaningfully contributed to improving the clarity and rigor of our work. We hope that our responses have adequately addressed the concerns raised.
>
> &nbsp;
>
> [1-1] Agarwal, Rishabh, et al. "Deep reinforcement learning at the edge of the statistical precipice." NeurIPS 2021.

---

### Decision · Program_Chairs · 2025-05-01

**Decision:**

Accept (poster)

**Comment:**

The work proposes a step of value function learning in between offline pretraining and online finetuning.

I was just surprised to see concerns about the statistical significance of the results after that this was the major reason for reject at a previous conference. However, the authors have addressed these concerns in the rebuttal and I expect their answers to be integrated in the final version. Overall, I agree with the reviewers and I think the paper is an interesting and relevant contribution.